# NEAR EXACT PRIVACY AMPLIFICATION FOR MATRIX MECHANISMS

**Christopher A. Choquette-Choo, Thomas Steinke, & Abhradeep Thakurta**
Google DeepMind
Mountain View, CA, 94043, USA
`{cchoquette, steinke, athakurta}@google.com`

**Arun Ganesh**
Google Research
Seattle, WA, 98103, USA
`arunganesh@google.com`

**Saminul Haque**
Department of Computer Science
Stanford University
Stanford, CA, 94305, USA
`saminulh@stanford.edu`

## ABSTRACT

We study the problem of computing the privacy parameters for DP machine learning when using privacy amplification via random batching and noise correlated across rounds via a correlation matrix $\mathbf{C}$ (i.e., the matrix mechanism). Past work on this problem either only applied to banded $\mathbf{C}$, or gave loose privacy parameters. In this work, we give a framework for computing near-exact privacy parameters for any lower-triangular, non-negative $\mathbf{C}$. Our framework allows us to optimize the correlation matrix $\mathbf{C}$ while accounting for amplification, whereas past work could not. Empirically, we show this lets us achieve smaller RMSE on prefix sums than the previous state-of-the-art (SOTA). We also show that we can improve on the SOTA performance on deep learning tasks. Our two main technical tools are (i) using Monte Carlo accounting to bypass composition, which was the main technical challenge for past work, and (ii) a "balls-in-bins" batching scheme that enables easy privacy analysis and implementation-wise is closer to shuffling (widely considered a practical random batching method) than Poisson sampling.

## 1 INTRODUCTION

Many recent works have improved the privacy-utility tradeoff of differentially private machine learning via improvements in *either* privacy amplification or how noise is correlated across rounds. Privacy amplification (Kasiviswanathan et al., 2008; Balle & Wang, 2018; Erlingsson et al., 2019a; Balle et al., 2020) uses randomness in data processing to improve the existing privacy guarantees. Correlated noise (Kairouz et al., 2021; Denisov et al., 2022; Choquette-Choo et al., 2022), or DP-FTRL, uses the matrix mechanism to add noise in DP training such that the noise added in one round is cancelled out in subsequent rounds. Specifically, if $\mathbf{z}$ is an i.i.d. Gaussian matrix, independent noise approaches, i.e., differentially private stochastic gradient descent (DP-SGD) clip gradients and add noise $\mathbf{z}[i, :]$ in the $i$th round of DP training; instead, DP-FTRL $(\mathbf{C}^{-1}\mathbf{z})[i, :]$ as the noise in the $i$th round, where $\mathbf{C}$ is a carefully chosen "correlation matrix".

Choquette-Choo et al. (2024a) showed that one could simultaneously obtain the benefits of privacy amplification and noise correlation, leading to a class of "banded"[1] $\mathbf{C}$ correlation matrices that Pareto dominated

---

[1] A square lower-triangular matrix is $b$-banded, if at most $b$ of the principal diagonals are non-zero.

either alone. However, their amplification analysis only applied to these specific banded matrices, whereas we may want to work with non-banded $\mathbf{C}$ due to utility or efficiency concerns (Dvijotham et al., 2024; McMahan et al., 2024). Choquette-Choo et al. (2024b) gives an amplification analysis that works with general (i.e., non-banded) $\mathbf{C}$ and is tight in the limit as $\varepsilon \to 0$, but for larger $\varepsilon$ values used commonly in DP machine learning, their analysis inherently introduces slack, and can greatly overestimate the true privacy parameter. Both of these works' limitations are due to a reliance on composition theorems. We propose an approach which bypasses composition and in turn avoids both of these limitations simultaneously:

*In this work, we use Monte Carlo accounting to enable nearly-exact privacy analysis and optimization of these amplified correlated noise mechanisms without any constraint on the structure of the correlation matrices. In turn, we are able to achieve smaller error for prefix sums than past work in various settings, and improve the state-of-the-art utility on deep learning tasks, even while using memory-efficient correlation matrices.*

This allows us to obtain significant amplification benefits for generic $\mathbf{C}$ matrices, which were shown to be better than constrained versions in both utility (Choquette-Choo et al., 2022) and memory (Dvijotham et al., 2024) but cannot yet be combined with privacy amplification in the practical regimes of interest for $\varepsilon$ in DP machine learning. In particular, we first consider the standard objective for correlated noise mechanisms, RMSE of prefix sums (as defined in e.g., Kairouz et al. (2021); Denisov et al. (2022); see Section 1.2), and find that our techniques lead to $\mathbf{C}$ with significant reduction in RMSE over the prior SOTA around $10\%$. When used to train models on the standard CIFAR-10 benchmark[2], the RMSE improvements translate to up to $1\%$ absolute accuracy improvements compared to the prior SOTA (Choquette-Choo et al., 2024a).

## 1.1 OUR CONTRIBUTIONS

### 1.1.1 ALGORITHMIC CONTRIBUTIONS

**Balls-in-bins minibatching:** We propose a more practical sampling scheme that (i) does not require random access to the entire dataset, (ii) achieves much better privacy amplification than the next-best alternative, shuffling, and (iii) enables efficient amplification analysis via Monte Carlo sampling which would otherwise be NP-hard to compute. We are able to implement a practical variant of this sampling scheme that enforces fixed batch sizes (which are required for compatability with modern ML techniques like XLA compilation) for our deep learning experiments.

**Near-exact analysis via Monte Carlo:** We recognize a common issue with prior amplification analysis of correlated noise mechanisms, which is their reliance on composition which led to slack for various reasons (see Section 1.2). We thus show how to use Monte Carlo accounting (Wang et al., 2023) which instead observes that approximate DP guarantees can be written as the expected value of some function of the privacy loss and $\varepsilon$ (see Section 2 for more details). Because it is easy to do Monte Carlo accounting for the whole matrix $\mathbf{C}\mathbf{x} + \mathbf{z}$, we can bypass the need for composition. In turn, we are able to obtain near-exact privacy analysis (only paying for the sampling error of Monte Carlo approximation, which can be made arbitrarily small) while supporting general $\mathbf{C}$, improving upon the weaknesses of both the past works.

**Optimizing C under amplification:** We, for the first time, show how to optimize $\mathbf{C}$ to minimize the RMSE under privacy amplification. This remediates a significant issue with prior work which required an expensive grid-search to check the amplified RMSE under all possible configurations (as this required post-hoc analysis after optimization of $\mathbf{C}$, where the optimization did not account for amplification). A crucial

---

[2]It is worth mentioning that accuracy of training on the CIFAR-10 benchmark with correlated noise has been highly optimized for over a long series of papers (e.g. (Choquette-Choo et al., 2022; 2024a)). So, any tangible and unconditional improvement is significant.

aspect of Monte Carlo accounting is that it allows one to calibrate the quality of the privacy guarantee based on the computation available, i.e., one can obtain a quick (and less rigorous) $\varepsilon$ parameter by using a small number of MC samples, which can be made rigorous (with appropriate failure probability $\delta$) by increasing the number of samples. So, while optimizing for the $\mathbf{C}$ matrices we use fewer MC samples, and when providing the final privacy guarantee with the optimized $\mathbf{C}$ matrix, we run it with a large number of samples to achieve appropriate convergence of the sampler.

### 1.1.2 EMPIRICAL EVALUATION

**RMSE analysis:** We first look at the standard objective for evaluating correlated noise mechanisms, RMSE on prefix sums (formally defined in Section 1.2). We compare the RMSE achieved by matrices amplified using our analysis, and by the previous SOTA of banded matrices and Poisson sampling of Choquette-Choo et al. (2024a). We demonstrate that because we can directly optimize the correlation matrix for the RMSE under amplification, we can achieve up to a 10% reduction in RMSE compared to the SOTA.

**Empirical evaluation on CIFAR-10:** We next use correlation matrices and $\sigma$ computed using our privacy analysis to train a VGG model on CIFAR-10. We again compare to the SOTA of Choquette-Choo et al. (2024a). We show that despite using a weaker sampling assumption, matrices amplified by balls-in-bins batching Pareto dominate the approach of (Choquette-Choo et al., 2024a), getting equal accuracy at smaller $\varepsilon$ and giving up to 1% absolute accuracy improvements at larger $\varepsilon$.

### 1.2 BACKGROUND AND PRIOR WORK

Here, we summarize prior work and highlight why our contributions above are significant.

**Privacy amplification:** There are several forms of privacy amplification, but the two most popular are sampling and shuffling. Poisson sampling forms batches by independently including each example with some probability. Exact analyses for DP-SGD with Poisson sampling are well-known, but Poisson sampling is generally considered impractical (Ponomareva et al., 2023). DP-SGD with shuffling uses multiple passes over a shuffled dataset with fixed batch size. Shuffling is considered more practical, but tight analyses of DP-SGD with shuffling remain elusive because of the dependence between examples.

**Correlated noise:** DP-SGD uses independent noise, i.e., if the gradients are the rows of a matrix $\mathbf{x}$, the gradients used in DP-SGD are the rows of $\mathbf{x} + \mathbf{z}$, where $\mathbf{z}$ is i.i.d. Gaussian noise the same shape as $\mathbf{x}$. DP-FTRL instead uses the correlated noise, i.e., uses rows of $\mathbf{x} + \mathbf{C}^{-1}\mathbf{z}$ as gradients, where $\mathbf{C}$ is a "correlation matrix." By post-processing this has the same privacy guarantees as the matrix mechanism $\mathbf{C}\mathbf{x} + \mathbf{z}$. Usually $\mathbf{C}$ is chosen to minimize some objective, the most common being the RMSE of prefix sums, given by $\left\|\mathbf{A}\mathbf{C}^{-1}\right\|_2 \cdot \sigma$ where $\mathbf{A}$ is the lower-triangular all-ones matrix and $\sigma$ is the noise multiplier needed for privacy. We give more background on privacy amplification and DP-FTRL in Appendix B.

**Challenges of combining privacy amplification and correlated noise:** The main hurdle is that privacy amplification analysis usually relies heavily on *composition*: instead of analyzing the entire training run directly, we compute a privacy guarantee for each round of training separately, and then combine these in a straightforward way to analyze the entire run. However, composition requires that both sampling and noise randomness be independent across rounds and of course noise correlations violates this. Thus, the only two prior works (Choquette-Choo et al., 2024a;b) combining privacy amplification and noise correlation leveraged reductions which significantly reduce the privacy amplification benefits.

**Amplified banded matrix factorization (Choquette-Choo et al., 2024a):** The matrix $\mathbf{C}$ is $b$-banded if only the first $b$ diagonals of $\mathbf{C}$ are non-zero. For $b$-banded $\mathbf{C}$, they observe that rows $i$ and $j$ of $\mathbf{C}\mathbf{x} + \mathbf{z}$ are independent if $|i - j| \geq b$. If each example is assigned an index $k \in \{0, 1, \ldots b - 1\}$ and then can only

be sampled in rounds $i$ where $i \pmod b \equiv k$, the resulting privacy guarantee can be reduced to standard DP-SGD but for $n/b$ rounds and batch size scaled up by $b$. However, there are two key limitations: (i) it constrains $\mathbf{C}$ limiting its expressivity, and (ii) its analysis does not exploit the fact that the assignment of batches to indices can be randomized. This latter issue becomes apparent when $b$ is sufficiently large, where the result collapses to one without any amplification despite this randomness.

**Conditional composition (Choquette-Choo et al., 2024b):** They instead show that these mechanisms have privacy guarantees upper bounded by a different independent noise mechanism, allowing the use of composition. Specifically, consider Poisson sampling with probability $p$. Effectively what they show is that we can compute a separate privacy guarantee for each row of $\mathbf{C}\mathbf{x} + \mathbf{z}$ using an inflated sampling probability $\widetilde{p} > p$ and compose these guarantees as if they were independent. This approach is more general than (Choquette-Choo et al., 2024a) because it applies to generic $\mathbf{C}$. However, $\widetilde{p}$ introduces slack in the analysis that gets worse as the noise multiplier decreases, such that the $\varepsilon$ computed in low-privacy regimes can be much worse than even the unamplified $\varepsilon$. This manifests practically in DP machine learning: most mechanisms obtain no benefits.

### 1.3 FUTURE DIRECTIONS

There are several interesting directions for extending the practicality of our work. One reason for our choice of balls-in-bins sampling is that it is more amenable to Monte Carlo accounting than Poisson sampling. Poisson sampling provides much stronger amplification than shuffling in most settings (Chua et al., 2024), so ignoring the practicality of Poisson sampling, enabling Monte Carlo accounting for correlated noise with Poisson sampling could thus enable even higher utility for private training. However, the fact that the corresponding mixture of Gaussians has $2^n$ modes is a challenging technical barrier to such a result.

The number of samples we need to verify a DP guarantee using Monte Carlo accounting and e.g. Bernstein's inequality scales as $1/\delta$, which may be prohibitive for small $\delta$. It is an interesting but challenging question to determine if a more careful sampler and concentration analysis can give substantially improved sample complexity for analyzing correlated noise mechanisms. Due to space constraints, we discuss other future directions in Appendix A.

## 2 PRIVACY LOSS DISTRIBUTIONS AND MONTE CARLO ACCOUNTING

We can define $(\varepsilon, \delta)$-DP in terms of the hockey-stick divergence. For any two distributions $P, Q$, their $\alpha$-hockey stick divergence for $\alpha \geq 0$ is given by $H_\alpha(P, Q) = \int_x \max\{P(x) - \alpha Q(x), 0\}$. A mechanism $\mathcal{M}$ is $(\varepsilon, \delta)$-DP if for any adjacent databases $D \sim D'$ (under the add-remove adjacency), we have $H_{e^\varepsilon}(\mathcal{M}(D), \mathcal{M}(D')) \leq \delta$. Suppose $P, Q$ are a *dominating pair* (Zhu et al., 2022) for $\mathcal{M}$ of interest; that is, for any $D \sim D', \alpha \geq 0$ we have $H_\alpha(\mathcal{M}(D), \mathcal{M}(D')) \leq H_\alpha(P, Q)$. Then, by sampling $X \sim P$ and computing $Y = \log P(X)/Q(X)$, we then say that the *privacy-loss distribution* (PLD) (Balle et al., 2018) of $P$ and $Q$ is the law of $Y$.

Now to prove $(\varepsilon, \delta)$-DP for $\mathcal{M}$, it suffices to show $H_{e^\varepsilon}(P, Q) \leq \delta$. The main observation in Monte Carlo accounting is that we can write $H_\alpha$ as an expectation over $Y$, $H_\alpha(P, Q) = \mathbb{E}_Y[\max\{1 - \alpha e^{-Y}, 0\}]$. Then, if exactly computing $H_\alpha(P, Q)$ is hard but sampling $Y$ is easy, we can estimate $\delta$ for some $\varepsilon$ by taking the average of sufficiently many samples of $\max\{1 - e^{\varepsilon-Y}, 0\}$. Namely, let $\hat{\delta}$ be the estimated $\delta$ value, i.e. the average over some samples of $\max\{1 - e^{\varepsilon-Y}, 0\}$. As a starting point, we can already claim $(\varepsilon, \hat{\delta})$-DP as an informal privacy statement. To make it formal, we can use the Estimate-Verify-Release framework of (Wang et al., 2023) (Algorithm 1): we fix a target $(\varepsilon, \delta)$-DP privacy guarantee ahead of time, and use Monte Carlo accounting to verify that $\mathcal{M}$ satisfies with $(\varepsilon, \delta)$-DP, and only run $\mathcal{M}$ if the verification succeeds.

The below theorem shows that Algorithm 1 can be made to incur only a constant blowup in $\delta$:

---

**Algorithm 1** Estimate-Verify-Release of (Wang et al., 2023)

    **Inputs:** Mechanism $\mathcal{M}$, dataset $D$, estimated privacy parameters $\varepsilon, \delta$.
1: Determine dominating pair $P, Q$ of $\mathcal{M}$, get $\hat{\delta}$, an estimate of $H_{e^\varepsilon}(P, Q)$.
2: **if** $\hat{\delta} \leq \delta$ **then**
3:     **return** $\mathcal{M}(D)$
4: **else**
5:     **return** $\perp$
6: **end if**

---

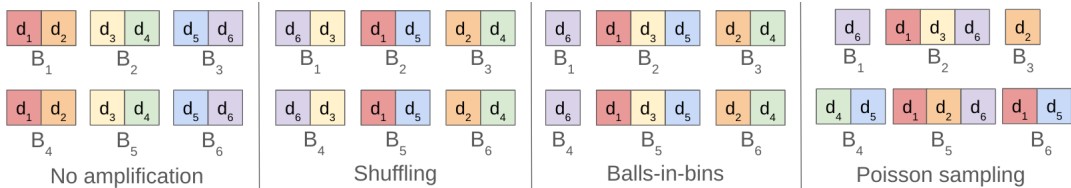

Figure 1: A comparison of the how different amplification methods might form batches. Here, we use $n = 6$ rounds, and form $b = 3$ batches per epoch for $E = 2$ epochs (visually represented as one row for each epoch). Note that shuffling uses fixed batch sizes, and all methods but Poisson sampling use the same batching across epochs and enforce exactly one participation per example per epoch.

**Theorem 2.1** (Theorem 9 of Wang et al. (2023)). *Suppose in Algorithm 1 that for any $P, Q$ such that $H_\varepsilon(P, Q) > \tau\delta$, $\hat{\delta} > \delta$ w.p. at least $1 - \tau\delta$. Then Algorithm 1 is $(\varepsilon, \tau\delta)$-DP.*

The following theorem (proven in Appendix E) can be used to derive the tail bound required for Theorem 2.1:

**Theorem 2.2.** *Suppose we use $s$ samples in computing $\hat{\delta}$ in Algorithm 1. Then for any $P, Q$ such that $H_\varepsilon(P, Q) \geq \tau\delta, \tau > 1$, we have $\mathbf{Pr}[\hat{\delta} < \delta] \leq \exp\left(-\frac{s(\tau-1)^2\delta}{8\tau/3-2/3}\right)$.*

## 3    BALLS-IN-BINS BATCHING

As we will see in the next section, applying Monte Carlo accounting to correlated noise mechanisms requires time linear in the number of possible participation patterns. For $n$-round DP-SGD with Poisson sampling, this is $2^n$. Shuffling reduces the number of possible participation patterns, and is generally considered a more practical variant, but is hard to do privacy analysis for because the participations of different examples are not independent like in Poisson sampling.

To alleviate the issues with both Poisson sampling and shuffling, we propose *balls-in-bins batching*. Informally, balls-in-bins batching forms batches using the balls-in-bins process, where examples are balls and batches are bins:

**Definition 3.1.** *In **balls-in-bins batching**, we form batches as follows. Let $b$ be the number of batches per epoch we wish to use and let $E$ be the number of epochs that we train. Also denote by $n = b \cdot E$ the total number of training iterations. For each example $d$, we independently include it in exactly one of $B_1, B_2, \ldots, B_b$ uniformly at random. We then iterate through the batches in round robin fashion, i.e. in iteration $i$ of our learning algorithm we use batch $B_{i \pmod{b}}$ (abusing notation to let e.g. $2b \pmod{b} = b$).*

In Figure 1 we give a visualization of balls-in-bins batching and other standard methods for forming batches. The following lemma (proven in Appendix C) shows that the following pair of distributions is a dominating

pair for the correlated noise mechanism $\mathbf{x} + \mathbf{C}^{-1}\mathbf{z}$ with balls-in-bins batching:

$$P = \frac{1}{b}\sum_{i=1}^{b} N(\mathbf{m}_i, \sigma^2 I), \text{ where } \mathbf{m}_i = \sum_{j=0}^{E-1} |\mathbf{C}|_{1:n, b \cdot j + i}, \qquad Q = N(0, \sigma^2 I),$$

where $|\mathbf{C}|$ is the matrix whose entries are the absolute values of corresponding entry in $\mathbf{C}$.

**Lemma 3.2** (Dimension-reduction for balls-in-bins batching). *Suppose $\mathbf{C} \in \mathbb{R}^{n \times n}$ is a lower-triangular matrix with non-negative entries. Then $P, Q$ (resp. $Q, P$) as defined above is a dominating pair for the correlated noise mechanism with balls-in-bins batching under the add (resp. remove) adjacency.*

For simplicity of presentation, we will focus on the add adjacency in the rest of the paper. It is easy to extend results to the remove adjacency, and for the add-and-remove adjacency we can simply take the worse of the two adjacencies' privacy guarantees (paying a factor of 2 in the failure probability of Theorem 2.2 by a union bound). We also remark that Lemma 3.2 (and any results in our paper for the add/remove adjacency) can be readily extended to the replace adjacency using recent results of Schuchardt et al. (2024).

**Advantages of balls-in-bins batching:** In addition to being more amenable for Monte Carlo analysis than shuffling or Poisson sampling, we believe balls-in-bins batching may be more practical than Poisson sampling for various reasons. First, one way to implement balls-in-bins batching to form $b$ batches is as follows: We sample $(c_1, c_2, \ldots, c_b)$ from the multinomial random variable with $|D|$ trials and $b$ equally likely outcomes. We shuffle the dataset once, and then we operate in multiple streaming passes over the shuffled dataset. In iteration $i$, we take the next $c_{i \pmod{b}}$ examples from the dataset stream.

Hence, up to the choice of variable batch size (which we will discuss how to make practical in Section I), balls-in-bins batching is no harder to implement than shuffling, which is generally considered to be a practical batch sampling method. As a function of requiring little overhead compared to shuffling, balls-in-bins batching avoids some of the practical issues with Poisson sampling. For example, shuffling and balls-in-bins are both memory-efficient if implemented in an offline manner as the shuffled dataset is the same size as the original dataset. In contrast, if we implement Poisson sampling in an offline manner, to store all batches formed by Poisson sampling we need to write each example $Bn/|D|$ times in expectation, which can be much larger than 1.

# 4 MONTE CARLO ACCOUNTING FOR CORRELATED NOISE MECHANISMS

## 4.1 CALIBRATING $\sigma$

Now, suppose we are given a lower triangular matrix $\mathbf{C} \in \mathbb{R}^{n \times n}$ with non-negative entries and we aim to find the minimum noise-multiplier $\sigma$ so that the balls-in-bins mechanism is $(\varepsilon, \delta')$-DP. This $\delta'$ will be slightly smaller than $\delta$ so that Algorithm 1 is $(\varepsilon, \delta)$-DP and outputs $\perp$ with very small probability. We approach this by drawing a sample of the dominating PLD and employing a bisection root-finding algorithm on $\sigma$.

To do this, we first recall the dominating distributions,

$$P_{\mathbf{C}, \sigma} = \frac{1}{b}\sum_{i=1}^{b} N(\mathbf{m}_i, \sigma^2 I), \text{ where } \mathbf{m}_i = \sum_{j=0}^{E-1} |\mathbf{C}|_{1:n, b \cdot j + i}, \qquad Q_\sigma = N(0, \sigma^2 I).$$

Hence we can sample $Y \sim PLD(P_{\mathbf{C}, \sigma}, Q_\sigma)$ by first sampling $X \sim P_{\mathbf{C}, \sigma}$ and computing $Y = \log\left(\frac{P_{\mathbf{C}, \sigma}}{Q_\sigma}(X)\right)$. Note that computing $Y$ is efficient since $P_{\mathbf{C}, \sigma}$ has only $b$ modes. To make the sampling independent of $\sigma$, we can instead sample $i \sim Uni([b])$ and $Z \sim N(0, I)$ then for any $\sigma$ compute

$Y = \log\left(\frac{P_{\mathbf{C},\sigma}}{Q_\sigma}(\mathbf{m}_i + \sigma \cdot Z)\right)$. This process is summarized in Algorithm 2, which uses the function $\hat{\delta}$ to estimate the $\delta$ at a fixed $\sigma$. The inner computation of $\hat{\delta}$ is easily parallelizable, as it is a simple average over a given sample size. On top of this, a bisection algorithm is run, which can converge[3] to within $\Delta$ additive error in only $O(\log(1/\Delta))$ many iterations because $\sigma$ is a scalar.

---

**Algorithm 2** Finding $\sigma$

---

**Inputs:** Target $(\varepsilon, \delta)$-DP guarantee, sample size $m$, matrix $\mathbf{C}$.

1: $i_1 \ldots i_m \overset{iid}{\sim} Uni([b])$
2: $Z_1, \ldots, Z_m \overset{iid}{\sim} N(0, I)$
3: Define $Y_j(\sigma) := \log\left(\frac{P_{\mathbf{C},\sigma}}{Q_\sigma}(\mathbf{m}_{i_j} + \sigma \cdot Z_j)\right)$
4: Define the function $\hat{\delta}$ as $\hat{\delta}(\sigma; \mathbf{C}, \varepsilon, i_{1:m}, Z_{1:m}) = \frac{1}{m}\sum_{j=1}^{m}(1 - \exp(\varepsilon - Y_j(\sigma)))_+$
5: $\sigma^\star \leftarrow$ solution to $\hat{\delta}(\sigma; \mathbf{C}, \varepsilon, i_{1:m}, Z_{1:m}) = \delta$ obtained by some 1-d bisection algorithm
6: **return** $\sigma^\star$

---

### 4.2 OPTIMIZING OVER MATRICES

In the previous section, we outlined the procedure Algorithm 2 for estimating the noise multiplier $\sigma$ for a given matrix $\mathbf{C}$. We now show that we can optimize $\mathbf{C}$ for a given utility metric that is a function of $\mathbf{C}, \sigma$. We will choose the utility metric to be the RMSE (root mean squared error) of prefix sums of $\mathbf{x}$ achieved by $\mathbf{x} + \mathbf{C}^{-1}\mathbf{z}$, which was also the metric optimized by (Choquette-Choo et al., 2022; 2024a). The RMSE is given by $\sigma \cdot \left\|\mathbf{A}^{-1}\mathbf{C}\right\|$, where $\left\|\cdot\right\|$ is the Frobenius norm and $\mathbf{A}$ is the all-ones lower-triangular matrix. While we focus on RMSE in this paper, our optimization framework is easily extended to any differentiable function of $\mathbf{C}$ and $\sigma$.

Algorithm 2 computes $\sigma$ (as a function of $\mathbf{C}$) using bisection, hence we cannot e.g. apply automatic differentiation to find partial derivatives of $\sigma$. However, via implicit differentiation we can still obtain gradients of the RMSE with respect to $\mathbf{C}$ using only quantities computable via automatic differentiation (see Appendix D for details). We then simply plug these gradients into an optimization algorithm; for simplicity, we use gradient descent. Finally, for efficiency, we choose to restrict to Toeplitz $\mathbf{C}$ to optimize over a lower dimensional space.

## 5 EXPERIMENTS

We implement Algorithm 2 for calibrating a noise multiplier for a given mechanism under balls-in-bins batching. In this section, we look at both the problem of minimizing RMSE and a deep learning setting, and compare results enabled by our accounting and optimization routines to past work. For all results we fix $\delta = 10^{-5}$, and for balls-in-bins batching we use Theorem 2.2 in conjunction with Theorem 2.1 to get a formal $(\varepsilon, 10^{-5})$-DP guarantee after choosing $\mathbf{C}$ and $\sigma$. See Appendix E for details.

---

[3]To show bisection search converges, if $\sigma_{max}$ is the $\sigma$ achieving the target DP guarantee without amplification (which we easily can compute using existing accounting libraries), $\hat{\delta}(0; \ldots) = 1 > \delta$ and $\hat{\delta}(\sigma_{max}; \ldots) < \delta$ (whp). In turn, $\hat{\delta}(\sigma; \ldots) = \delta$ for some $\sigma \in [0, \sigma_{max}]$ and bisection converges to this point by continuity of $\hat{\delta}$.

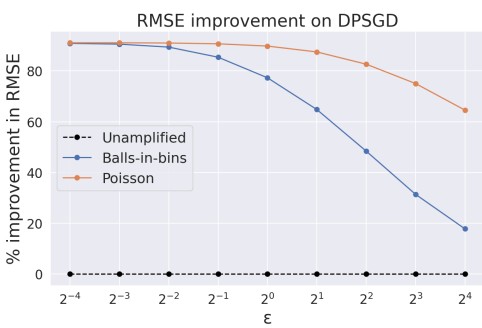
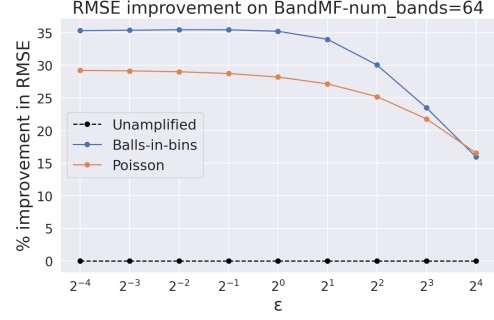

(a) Improvement in RMSE over unamplified DP-SGD due to different amplification schemes.

(b) Improvement in RMSE over unamplified 64-banded matrix factorization due to different amplification schemes.

Figure 2: Comparisons using fixed $\mathbf{C}$.

## 5.1 RMSE RESULTS

### 5.1.1 ANALYSIS OF THE AMPLIFICATION SCHEMES

We first focus on the RMSE metric $\sigma \cdot \|\mathbf{A}^{-1}\mathbf{C}\|$. We compare our balls-in-bins batching analysis, Poisson sampling (using the analysis of (Choquette-Choo et al., 2024a) for $b$-banded matrices where $b > 1$, and standard PLD accounting for DP-SGD), and no amplification (i.e., we do multiple passes using a fixed batch size over a dataset). To simplify presentation in this section we will restrict to the setting with 2048 total iterations and 16 epochs, i.e. 128 iterations per epoch or $B = |D|/128$. In Appendix F, we provide results for other settings of the number of epochs and number of iterations per epoch; while the exact numbers differ in other settings, the high-level conclusions hold in all settings we tested.

**Amplification of DP-SGD:** In Fig. 2a, we plot the percentage improvement in RMSE due to balls-in-bins and Poisson sampling over the unamplified RMSE of DP-SGD, i.e. $\mathbf{C} = \mathbf{I}$. We observe that the improvement from the two sampling schemes is similar for smaller $\varepsilon$ and Poisson performs substantially better than balls-in-bins for larger $\varepsilon$. Since balls-in-bins resembles shuffling, this mirrors the results of Chua et al. (2024).

**Amplification of banded matrix factorization:** We next consider non-identity choices of $\mathbf{C}$. In particular, we look at $b$-banded matrices where $b > 1$. Recall that a weakness of the sampling scheme of Choquette-Choo et al. (2024a) is that the amount of randomness in the sampling scheme, i.e. the degree of benefit from amplification, decreases as the number of bands $b$ in the matrix increases. In contrast, the randomness in sampling from balls-in-bins is independent of the matrix structure. We then may expect that balls-in-bins amplification outperforms the amplification of Choquette-Choo et al. (2024a). We verify this in Fig. 2b: We again plot the percentage reduction in RMSE due to amplification but for a 64-banded matrix instead of $\mathbf{C} = \mathbf{I}$. As predicted, due to using a higher-randomness sampling scheme, balls-in-bins consistently outperforms Poisson sampling until $\varepsilon = 16$.

While the previous experiment shows that balls-in-bins is preferable when using a fixed large number of bands, a better strategy is to choose the number of bands $b$ to minimize the RMSE rather than fix $b$ in advance. In Fig. 3a, we reproduce Figure 2b but instead of fixing the number of bands $b$ in advance, we sweep $b \in \{1, 2, 4, 8, 16, 32, 64, 128, 256\}$ and for balls-in-bins batching pick the value of $b$ which minimizes the RMSE. For banded matrices and Poisson sampling, we do the same but restrict to $b \leq 64$, i.e. the regime where Choquette-Choo et al. (2024a)'s result gives a non-zero amount of amplification. For the unamplified baseline, there is no benefit to reducing the number of bands $b$ so we pick $b = 256$.

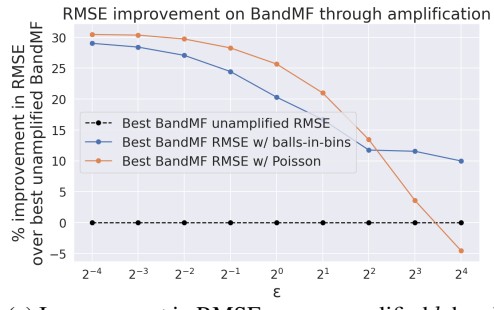

(a) Improvement in RMSE over unamplified $b$-banded matrix factorization due to different amplification schemes. All curves optimize the choice of $b$ at each point. Best choices of $b$ are plotted in Appendix H.

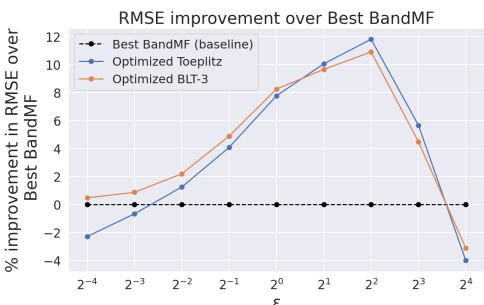

(b) Improvement from optimizing $\mathbf{C}$ using objective accounting for amplification.

Figure 3: Comparisons for variable $\mathbf{C}$.

For $\varepsilon \leq 4$, the best RMSE under Poisson sampling slightly outperforms balls-in-bins batching and for $\varepsilon \geq 8$, we see a deterioration in the RMSE improvements from Poisson. At $\varepsilon = 16$, we see that it is better to forgo amplification and use the unamplified baseline. In contrast, because the distinguishing problem for balls-in-bins for a fixed $\mathbf{C}$ is always strictly harder than the unamplified baseline, the best $\mathbf{C}$ combined with balls-in-bins batching always strictly improves over the unamplified baseline.

### 5.1.2 IMPROVEMENTS DUE TO OPTIMIZATION

We now look at the benefits of using our optimization procedure to choose $\mathbf{C}$. We compare two variants of our optimization scheme, one which optimizes over general Toeplitz matrices, and another which optimizes over the BLT matrix family defined by Dvijotham et al. (2024). We use the 3-buffer variant of BLTs, i.e. they can be specified using 6 parameters and only require a memory overhead of 3 times the model size. In Figure 3b we plot the improvement in RMSE for both these variants relative to the best $b$-banded $\mathbf{C}$ amplified using the sampling scheme and analysis of Choquette-Choo et al. (2024a).

We observe that for most $\varepsilon$ values, our optimization procedure for $\mathbf{C}$ can give a reduction in RMSE compared to the previous state-of-the-art of Choquette-Choo et al. (2024a). For extreme values of $\varepsilon$, the banded baseline still outperforms our optimized matrices. We note that despite BLT matrices being a subset of Toeplitz matrices, our optimized BLTs matrices sometimes outperform our optimized Toeplitz matrices. We believe this is because BLTs lie in a lower-dimensional space which may make it easier to optimize over them, i.e. the suboptimality of our solution to the optimization problem over BLTs may be smaller than the suboptimality of our solution to the problem over Toeplitz matrices.

We also conducted experiments to understand the scalability of our optimization procedure. Due to space constraints, we defer these to Appendix G.

### 5.2 CIFAR RESULTS

We next compare different choices of the correlation matrix $\mathbf{C}$ and amplification schemes in a deep learning setting. We replicate the CIFAR10 image recognition setting considered by (Choquette-Choo et al., 2024a): we also use the same VGG model, 20 epochs of 100 iterations with batch size 500, and momentum of 0.95 and a learning rate cooldown from $\eta$ to $\eta/20$ across iterations 500 to 2000. We vary $\varepsilon \in \{0.5, 1.0, 2.0, 4.0, 8.0\}$. We fix the clip norm to 1.0 and tune the learning rate separately for each com-

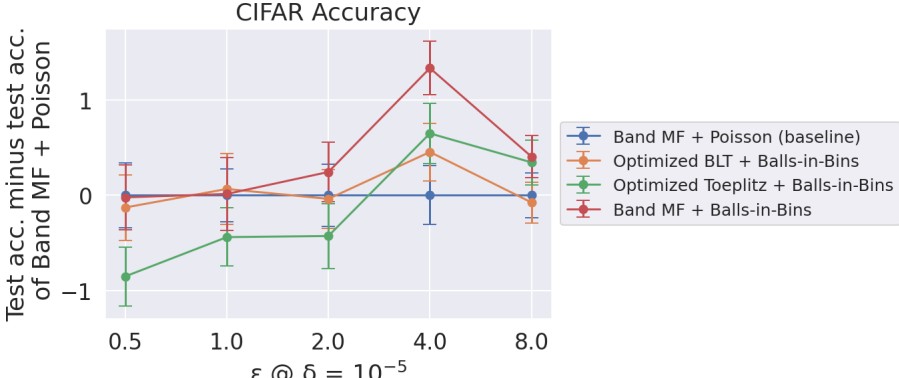

Figure 4: Comparison of accuracy on CIFAR of different correlated noise strategies and amplification methods, relative to the baseline of banded $\mathbf{C}$ + Poisson sampling, with 95% confidence intervals over 100 trials. We give a non-normalized plot in Appendix H.

bination of $\varepsilon$ and correlation matrix/amplification method, and then report the average of 100 training runs for each combination. The different correlation matrix choices and amplification methods we consider are:

- Our baseline is the SOTA of banded matrices of Choquette-Choo et al. (2024a) using their Poisson sampling-based amplification scheme. We also consider banded matrices and balls-in-bins batching. Following Choquette-Choo et al. (2024a), for each choice of the number of bands $b$, we optimize $\mathbf{C}$ without accounting for amplification. Then for each amplification method, we use the choice of $b$ in each setting of $\varepsilon$ that gives the lowest RMSE under amplification.
  - We do not separately consider multi-epoch MEMF of Choquette-Choo et al. (2022) or DP-SGD with Poisson sampling as they are subsumed by this approach.
- BLT matrices of Dvijotham et al. (2024). We restrict to 4 buffers and use our optimization framework to optimize $\mathbf{C}$ for the RMSE under balls-in-bins batching for each value of $\varepsilon$.
- General Toeplitz lower-triangular matrices. Again we use our optimization framework to optimize $\mathbf{C}$ for the RMSE under balls-in-bins batching for each value of $\varepsilon$.

For all methods, in the implementation we shuffle the dataset and use a fixed batch size, but calculate the noise multiplier assuming the corresponding amplification method was properly implemented. This is a standard practice for simplifying implementations in the literature (e.g., Choquette-Choo et al. (2024a)). In Appendix I, we discuss the pitfalls of this practice, and do a training run where we actually implement a practical variant of balls-in-bins batching that also uses fixed batch sizes for gradient computations.

In Fig. 4, we plot the test accuracy achieved by each combination of correlation matrix and amplification analysis. We can conclude the following:

- Banded matrix factorization + balls-in-bins is always at least as good as the baseline, and sometimes gives up to 1% accuracy improvements. As argued earlier, the baseline assumes Poisson sampling which is a stronger/less practical assumption.
- Our ability to optimize BLT matrices makes them also always at least as good as the baseline, but with smaller improvements than the previous bullet. The baseline uses up to 16 bands depending on $\varepsilon$, so BLTs are also up to 4x more memory efficient than the baseline.
- Optimized Toeplitz matrices are incomparable to the baseline; we believe their weak performance relative to the other methods is due to the difficulty of the optimization problem.

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

## A    MORE FUTURE DIRECTIONS

To be able to use Monte Carlo sampling to analyze the privacy amplification of the balls-in-bins sampling scheme, we needed to first provide a PLD-dominating pair of distributions. In Lemma 3.2, we obtain this pair by taking the modes of $P$ to be sums of certain columns of $|\mathbf{C}|$. This is potentially too loose, and we may hope to instead use sums of columns of $\mathbf{C}$, thereby reducing the norm of each mode. In the unamplified setting, this loosening of the assumption is possible whenever $\mathbf{C}[:, i] \cdot \mathbf{C}[:, j]$ for any pair of rounds $i, j$ a user can participate in. However, amplification through sampling complicates this, and standard techniques for reducing dimensionality and adaptivity don't work anymore, see Appendix J for an example.

While we were able to implement a practical variant of balls-in-bins batching that was compatible with XLA compilation, we only used our variant to train a small-scale model. We leave it to future work to implement balls-in-bins batching at larger scales, and in particular compare the scalability of balls-in-bins to that of unamplified training and Poisson sampling.

## B    MORE BACKGROUND ON PRIVACY AMPLIFICATION AND CORRELATED NOISE

**Privacy amplification:** Though randomness in data processing reduces the variance of noise required, the main challenge with privacy amplification is giving tight privacy analysis of the amplified mechanism, which for DP-SGD usually requires computing a divergence between two mixtures of Gaussians. There are several forms of privacy amplification, but the two most popular are sampling (Abadi et al., 2016; Balle et al., 2018) and shuffling (Balle & Wang, 2018; Erlingsson et al., 2019b;a; Feldman et al., 2022).

Privacy amplification by (Poisson) sampling assumes the batches are formed by including each element independently with some probability $p$. Because the sampling randomness is independent across rounds, tight privacy analyses of DP training with independent noise (DP-SGD) are well-known (Koskela et al., 2021), and supported by open-source libraries (DP Team, 2022). However, Poisson sampling is often impractical to implement, especially when working with larger datasets (Ponomareva et al., 2023).

Privacy amplification by shuffling in the context of DP-SGD usually assumes the batches are formed by shuffling the dataset (a single time) and then forming fixed-size batches using the shuffled order. In some limited settings its privacy analysis is well-understood (Feldman et al., 2022), but in the setting of DP-SGD, for example, tight privacy analyses remain elusive. Technically, the difficulty is that the participations of different examples are dependent, unlike with Poisson sampling. In addition to being challenging to analyze, Chua et al. (2024) demonstrated that shuffling often gives much smaller improvements in privacy compared to sampling. Despite these difficulties, as Chua et al. (2024) note, shuffling is used much more widely in practice since it can be implemented, e.g., in a single pre-processing pass over the dataset, whereas sampling requires random access to potentially very large datasets which is often infeasible.

**Correlated noise:** Kairouz et al. (2021) proposed DP-FTRL, the first DP training algorithm with correlated noise, where the update at each step are the rows of $\mathbf{x} + \mathbf{z}_{tree}$ where $\mathbf{z}_{tree}$ is sampled using the binary tree mechanism of (Dwork et al., 2010). Denisov et al. (2022) showed this was a special case of the updates being rows of $\mathbf{x} + \mathbf{C}^{-1}\mathbf{z}$, giving a more expressive definition for $\mathbf{C}^{-1}$ called the "correlation matrix". They showed these matrices could be optimized via a convex optimization program to maximize noise cancellation (i.e., minimize total noise variance). Choquette-Choo et al. (2022) proposed a "multi-participation" generalization, leading to the first correlated noise DP training algorithm to (substantially) outperform DP-SGD with practical $\varepsilon$'s. Choquette-Choo et al. (2024a) showed that correlated noise algorithms Pareto-dominated DP-SGD in privacy-utility tradeoffs; Choquette-Choo et al. (2023) showed that these algorithms provably outperformed DP-SGD.

For simplicity, these works assume $\mathbf{C}$ is non-negative and lower-triangular. The privacy guarantee of correlated noise is the same as that of the matrix mechanism $\mathbf{Cx} + \mathbf{z}$ by post-processing, hence based on how batches are formed we can usually give privacy guarantees in terms of an appropriate norm of $\mathbf{C}$. To optimize the matrix $\mathbf{C}$, the convex program usually minimizes the root-mean-squared-error (RMSE) of the prefix sums of $\mathbf{x}$, given by $\left\|\mathbf{AC}^{-1}\right\|_2 \cdot \sigma$ where $\mathbf{A}$ is the lower-triangular all-ones matrix and $\sigma$ is the noise multiplier needed for $\mathbf{Cx} + \mathbf{z}$ to satisfy the target privacy guarantee.

## C  PROOF OF LEMMA 3.2

We use Lemma 4.5 of (Choquette-Choo et al., 2024b), restated here for convenience:

**Lemma C.1.** *Let* $\mathbf{c}_1, \ldots, \mathbf{c}_k \in \mathbb{R}^{n \times p}$. *Let* $\mathbf{c}'_1, \ldots, \mathbf{c}'_k \in \mathbb{R}^n$ *be such that* $\|\mathbf{c}_i[j,:]\|_2 \leq \mathbf{c}'_i(j)$ *for all* $i, j$. *Then letting*

$$P = N(0, \sigma^2 \mathbb{I}_{(n\times p)\times(n\times p)}), Q = \sum_i p_i N(\mathbf{c}_i, \sigma^2 \mathbb{I}_{(n\times p)\times(n\times p)})$$

$$P' = N(0, \sigma^2 \mathbb{I}_{n\times n}), Q' = \sum_i p_i N(\mathbf{c}'_i, \sigma^2 \mathbb{I}_{n\times n}),$$

*for all* $\alpha$ *we have* $H_\alpha(P, Q) \leq H_\alpha(P', Q')$. *Furthermore, this holds even if the $j$th row of each $\mathbf{c}_i$ is chosen as a function of the first $j-1$ rows of $P, Q$ (subject to $\|\mathbf{c}_i[j,:]\|_2 \leq \mathbf{c}'_i(j)$) while $\mathbf{c}'_i$ remain fixed.*

*Proof.* We give the proof for the add adjacency. Proving $P, Q$ is a dominating pair for the add adjacency implies $Q, P$ is a dominating pair for the remove adjacency by Lemma 29 of (Zhu et al., 2022).

Because each user is assigned to their batch independently, we can assume without loss of generality that contributions from all users other than the differing user are always 0. In more detail, by post-processing, we can assume that we release the contributions to the input matrix of all examples except the differing user's. Let $\mathbf{x}$ be these contributions, and $\mathbf{x}'$ be $\mathbf{x}$ plus the contribution of the differing user. Then distinguishing $\mathbf{Cx} + \mathbf{z}$ and $\mathbf{Cx}' + \mathbf{z}$ is equivalent to distinguishing $\mathbf{z}$ and $\mathbf{C}(\mathbf{x}' - \mathbf{x}) + \mathbf{z}$, exactly the setting where all users except the differing user only contribute 0.

Now, for the input matrix $\mathbf{x} \in \mathbb{R}^{n \times p}$, the rows have at most unit norm in the entries where the differing user participates and 0 everywhere else. If the differing user is assigned to the $i$th batch, it is immediate by triangle inequality that $\|(\mathbf{Cx})_{k,:}\| \leq \sum_{j=0}^{E-1} |\mathbf{C}|_{k, b\cdot j + i}$. Because every user participates in each batch with probability $\frac{1}{b}$, we can apply Lemma 4.5 of Choquette-Choo et al. (2024b) to exactly get that the correlated noise mechanism with balls-with-bins batching is dominated by the pair $P, Q$. $\qquad\square$

## D  FINDING THE DERIVATIVE OF RMSE

By chain rule:

$$\frac{\partial(\sigma \cdot \|\mathbf{A}^{-1}\mathbf{C}\|)}{\partial \mathbf{C}} = \sigma \cdot \frac{\partial \|\mathbf{A}^{-1}\mathbf{C}\|}{\partial \mathbf{C}} + \|\mathbf{A}^{-1}\mathbf{C}\| \cdot \frac{\partial \sigma}{\partial \mathbf{C}}.$$

Let $\hat{\delta}$ denote the function as defined in Algorithm 2, i.e. the function that takes a given $\varepsilon, \sigma, \mathbf{C}$ and set of Monte Carlo samples, and outputs the corresponding estimated $\delta$ value. Given a set of Monte Carlo samples, we want to take a gradient step while preserving $\hat{\delta} = \delta$. Taking derivative of both sides of this constraint with respect to $\mathbf{C}$:

$$\frac{\partial\hat{\delta}}{\partial\mathbf{C}} + \frac{d\hat{\delta}}{d\sigma}\cdot\frac{\partial\sigma}{\partial\mathbf{C}} = 0 \implies \frac{\partial\sigma}{\partial\mathbf{C}} = -\left(\frac{\partial\hat{\delta}}{\partial\mathbf{C}}\right) \Big/ \left(\frac{d\hat{\delta}}{d\sigma}\right)$$

So to optimize the RMSE w.r.t. $\mathbf{C}$ we can now use Algorithm 2 to find $\sigma$ for a given $\mathbf{C}$, and then apply gradient descent using the gradient

$$\frac{\partial(\sigma\cdot\|\mathbf{A}^{-1}\mathbf{C}\|)}{\partial\mathbf{C}} = \sigma\cdot\frac{\partial\|\mathbf{A}^{-1}\mathbf{C}\|}{\partial\mathbf{C}} - \|\mathbf{A}^{-1}\mathbf{C}\|\left(\frac{\partial\hat{\delta}}{\partial\mathbf{C}}\right) \Big/ \left(\frac{d\hat{\delta}}{d\sigma}\right).$$

Since $\hat{\delta}$ is computed using only differentiable functions of $\mathbf{C}$ and $\sigma$ (i.e., not via bisection), all partial derivatives in the above expression can be evaluated at the given $\mathbf{C}, \sigma$ pair using automatic differentiation.

## E    CONFIDENCE INTERVALS FOR MONTE CARLO ACCOUNTING

In this section we prove Theorem 2.2, which allows one to derive the required tail bounds on the error of $\hat{\delta}$ in Theorem 2.1.

*Proof of Theorem 2.2.* If $H_\varepsilon(P,Q) > \tau\delta$, then we can express $\hat{\delta} = \frac{1}{s}\sum_{i=1}^{s}\hat{\delta}_i$, where each $\hat{\delta}_i$ is an i.i.d. random variable in the range $[0,1]$ with mean $\mu \geq \tau\delta$. By the Bhatia-Davis inequality, we have $\mathbf{Var}\left(\hat{\delta}_i\right) \leq (1-\mu)\mu$. So $\mathbb{E}[\hat{\delta}_i^2] = \mathbb{E}[\hat{\delta}_i^2]^2 + \mathbf{Var}\left(\hat{\delta}_i\right) = \mu^2 + \mathbf{Var}\left(\hat{\delta}_i\right) \leq \mu$.

Now we can apply Bernstein's inequality to get that:

$$\mathbf{Pr}[\hat{\delta} < \mu - t] \leq \exp\left(-\frac{st^2/2}{\mathbb{E}[\hat{\delta}_1^2] + t/3}\right) \leq \exp\left(-\frac{st^2/2}{\mu + t/3}\right).$$

Setting $t = \mu - \delta$:

$$\mathbf{Pr}[\hat{\delta} < \delta] \leq \exp\left(-\frac{s(\mu-\delta)^2/2}{4\mu/3 - \delta/3}\right).$$

Since $\mu \geq \tau\delta > \delta$, we have:

$$\frac{d}{d\mu}\frac{s(\mu-\delta)^2/2}{4\mu/3 - \delta/3} = \frac{3s(\mu-\delta)(2\mu-\delta)}{(4\mu-3\delta)^2} > 0,$$

i.e. the bound $\mathbf{Pr}[\hat{\delta} < \delta]$ is decreasing in $\mu$, so it is minimized by setting $\mu = \tau\delta$. Plugging this value of $\mu$ in gives as desired:

$$\mathbf{Pr}[\hat{\delta} < \delta] \leq \exp\left(-\frac{s(\tau-1)^2\delta}{8\tau/3 - 2/3}\right).$$

$\square$

For our empirical results, we use $\delta = 8 \cdot 10^{-6}, \tau = 1.25, s = 10^8$. For this set of values, Theorem 2.2 gives a failure probability less than $7.2 \cdot 10^{-9}$. We use Algorithm 1 to verify the DP guarantee for both the add and remove adjacencies, but by a union bound the failure probability increases to at most $1.5 \cdot 10^{-8}$ once accounting for both adjacencies, much smaller than $\tau\delta = 10^{-5}$. So, we can apply Theorem 2.1 to show $(\varepsilon, 10^{-5})$-DP for all our empirical results.

## F   RMSE PLOTS FOR OTHER SETTINGS

### F.1   VARYING SAMPLING PROBABILITY

In Figure 5 we compare Poisson sampling and balls-in-bins batching. We fix 128 rounds of training, and fix a target DP guarantee of $(1, 10^{-5})$-DP. We vary the sampling probability $p$ and compare the RMSE achieved by the best banded matrix with either balls-in-bins sampling + our amplification analysis or the sampling scheme and analysis of Choquette-Choo et al. (2024a). For balls-in-bins, we treat $1/b$ where $b$ is the iterations per epoch as the sampling probability for the purpose of plotting, i.e. at each point on the x-axis both algorithms have the same expected batch size.

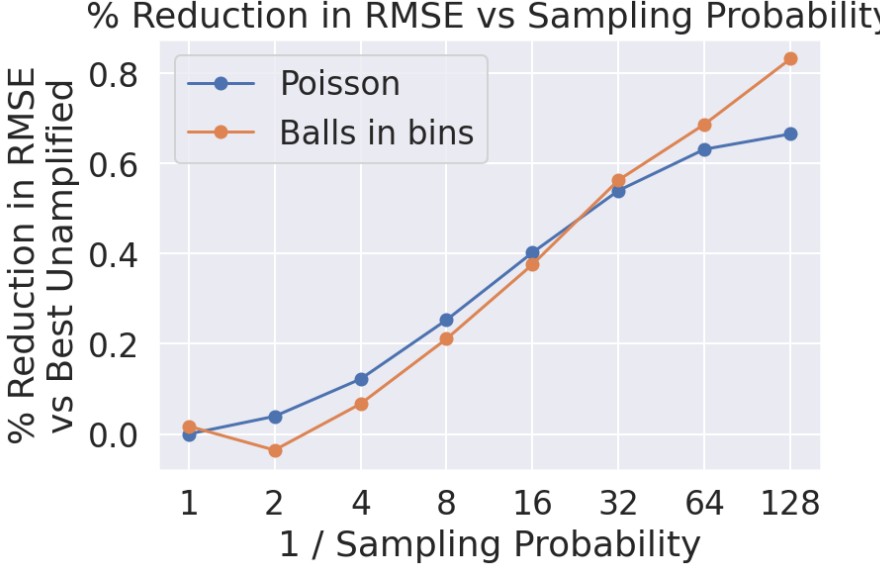

Figure 5: Comparison of improvement due to as a function of sampling probability.

### F.2   GRID OF EPOCHS AND ITERATIONS

Here we plot the RMSE improvements through amplification and optimization for number of epochs in $\{8, 16, 32\}$ and iterations per epoch in $\{32, 64, 128\}$.

### F.3 Number of epochs $= 8$ and iterations per epoch $= 32$

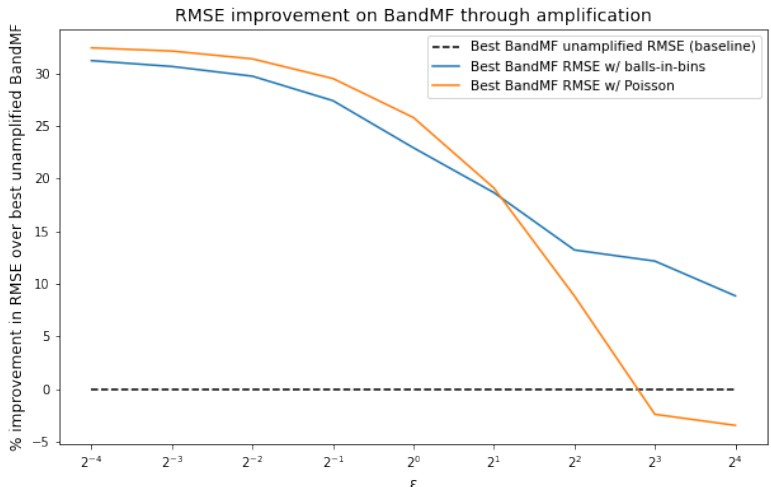

Figure 6: Improvement in RMSE over unamplified $b$-banded matrix factorization due to different amplification schemes. All curves optimize the choice of $b$ at each point.

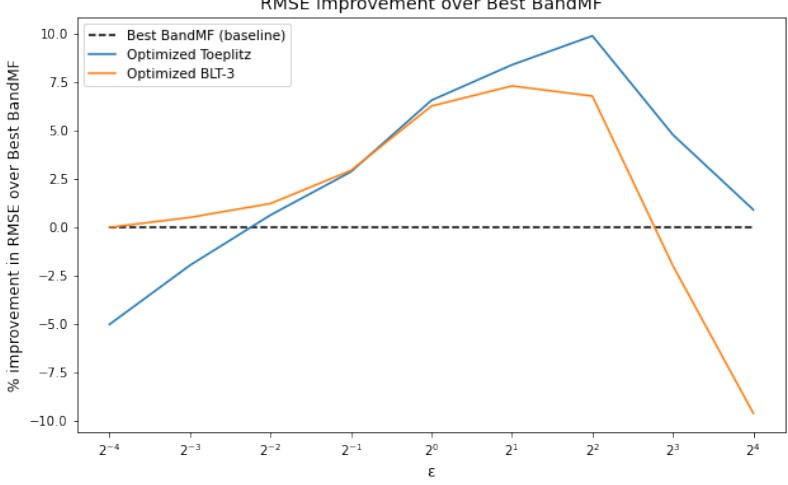

Figure 7: Improvement from optimizing $\mathbf{C}$ using objective accounting for amplification.

## F.4 NUMBER OF EPOCHS $= 8$ AND ITERATIONS PER EPOCH $= 64$

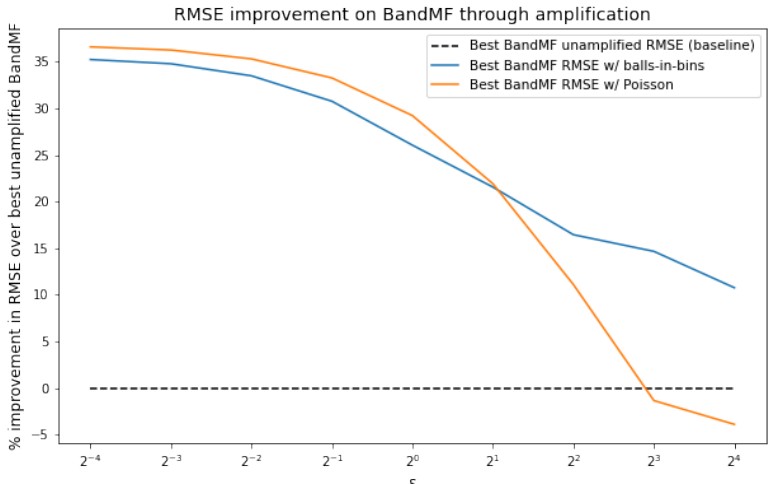

Figure 8: Improvement in RMSE over unamplified $b$-banded matrix factorization due to different amplification schemes. All curves optimize the choice of $b$ at each point.

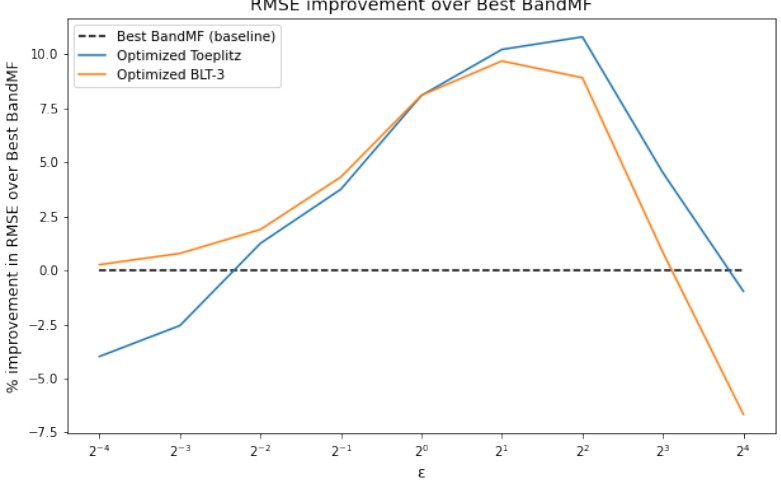

Figure 9: Improvement from optimizing $\mathbf{C}$ using objective accounting for amplification.

### F.5 NUMBER OF EPOCHS = 8 AND ITERATIONS PER EPOCH = 128

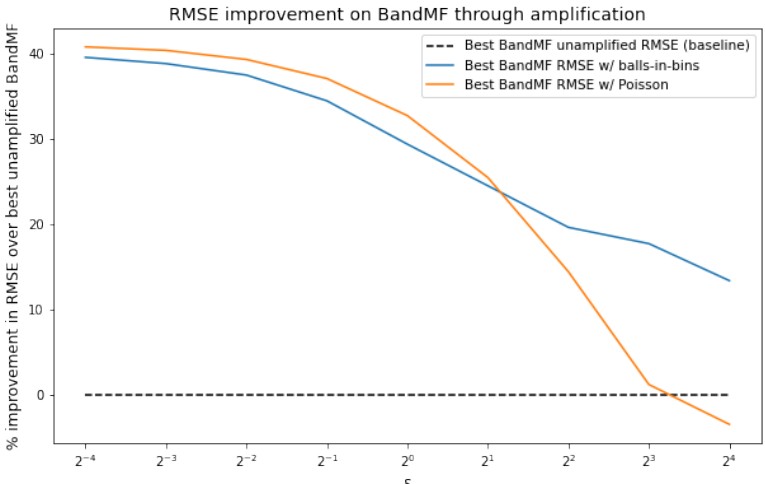

Figure 10: Improvement in RMSE over unamplified $b$-banded matrix factorization due to different amplification schemes. All curves optimize the choice of $b$ at each point.

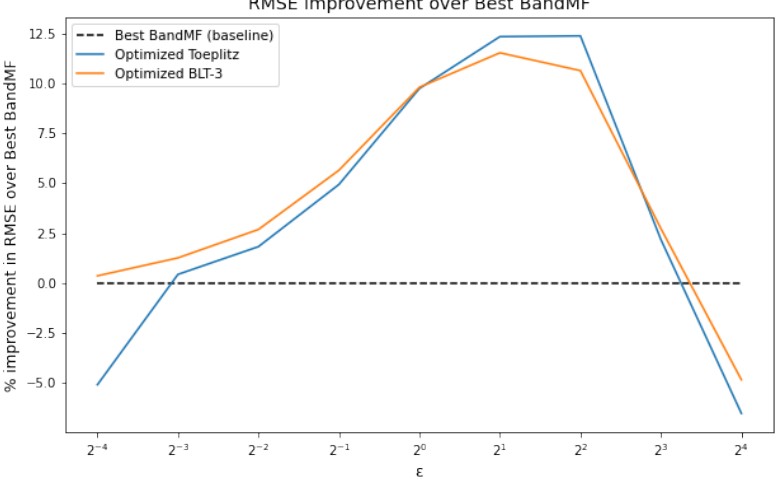

Figure 11: Improvement from optimizing $\mathbf{C}$ using objective accounting for amplification.

F.6 NUMBER OF EPOCHS $= 16$ AND ITERATIONS PER EPOCH $= 32$

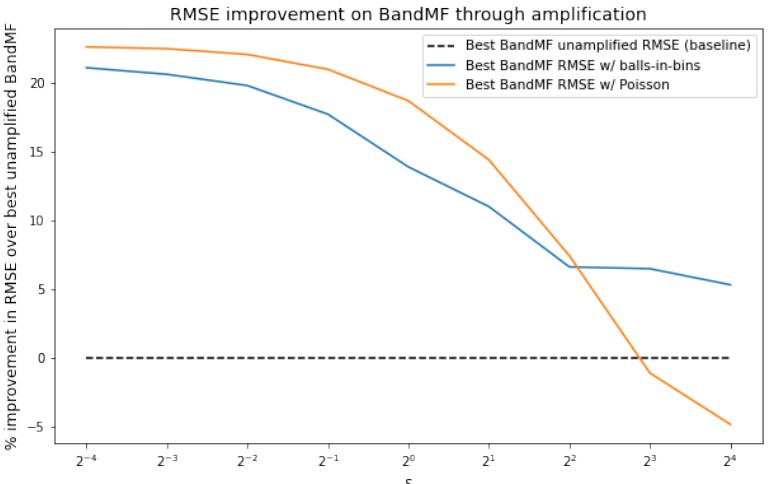

Figure 12: Improvement in RMSE over unamplified $b$-banded matrix factorization due to different amplification schemes. All curves optimize the choice of $b$ at each point.

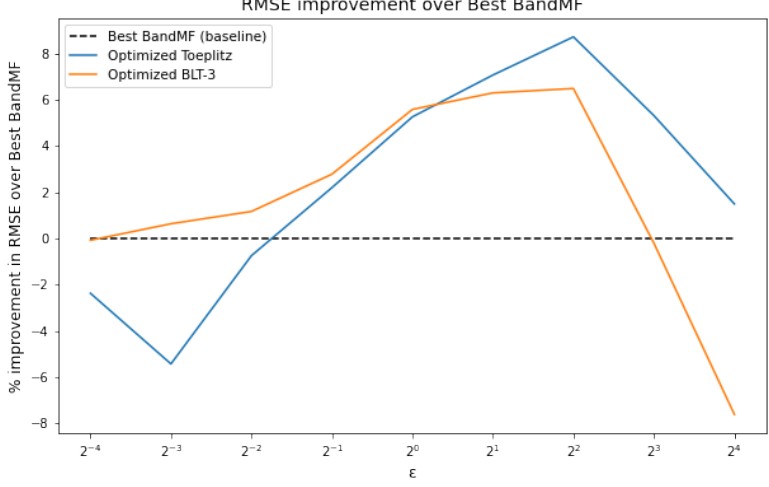

Figure 13: Improvement from optimizing $\mathbf{C}$ using objective accounting for amplification.

### F.7 NUMBER OF EPOCHS $= 16$ AND ITERATIONS PER EPOCH $= 64$

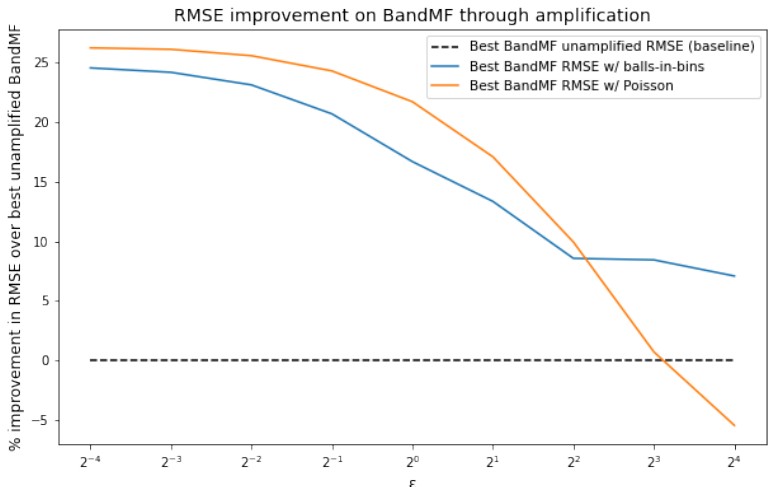

Figure 14: Improvement in RMSE over unamplified $b$-banded matrix factorization due to different amplification schemes. All curves optimize the choice of $b$ at each point.

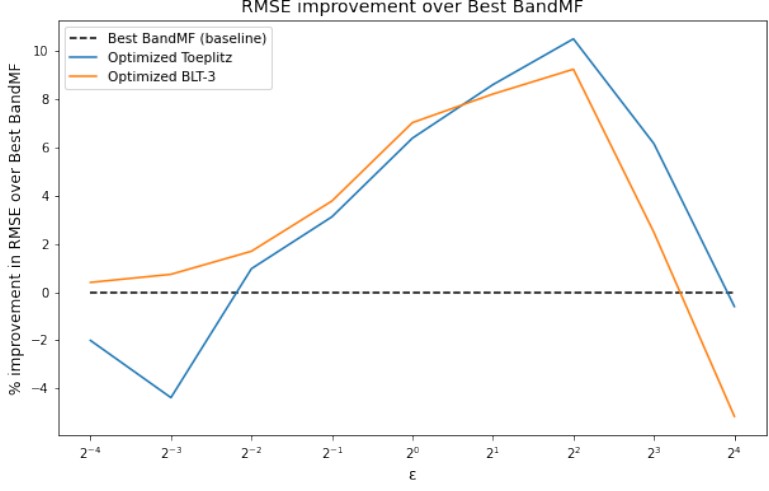

Figure 15: Improvement from optimizing $\mathbf{C}$ using objective accounting for amplification.

### F.8 NUMBER OF EPOCHS $= 16$ AND ITERATIONS PER EPOCH $= 128$

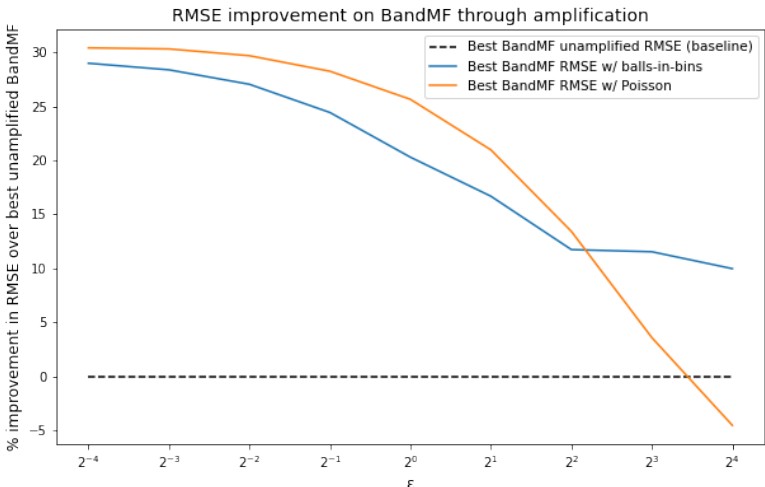

Figure 16: Improvement in RMSE over unamplified $b$-banded matrix factorization due to different amplification schemes. All curves optimize the choice of $b$ at each point.

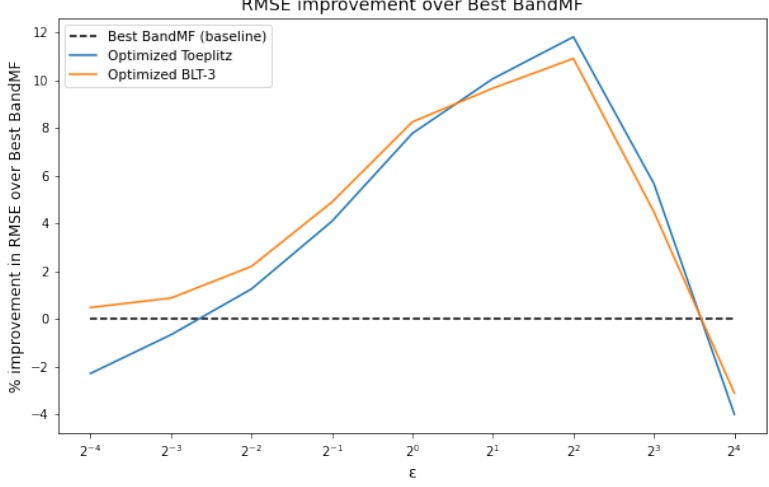

Figure 17: Improvement from optimizing $\mathbf{C}$ using objective accounting for amplification.

### F.9 NUMBER OF EPOCHS = 32 AND ITERATIONS PER EPOCH = 32

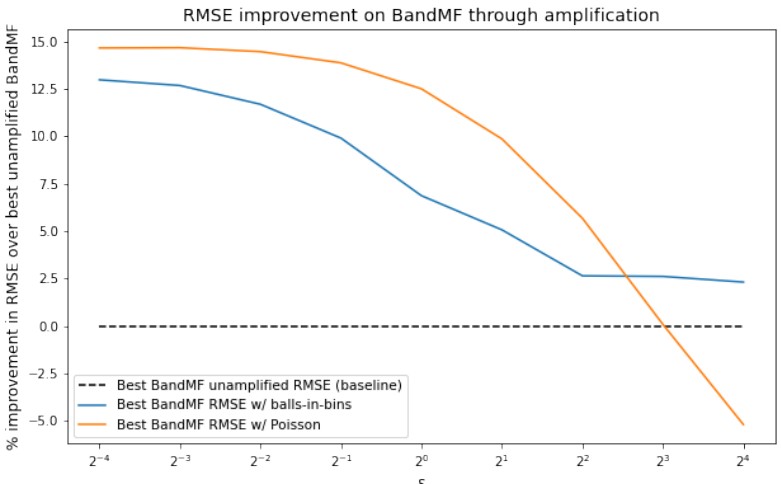

Figure 18: Improvement in RMSE over unamplified $b$-banded matrix factorization due to different amplification schemes. All curves optimize the choice of $b$ at each point.

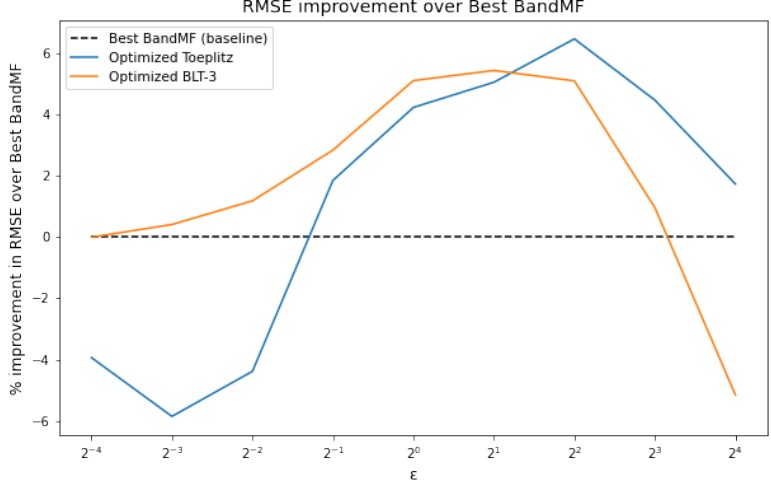

Figure 19: Improvement from optimizing $\mathbf{C}$ using objective accounting for amplification.

### F.10 NUMBER OF EPOCHS = 32 AND ITERATIONS PER EPOCH = 64

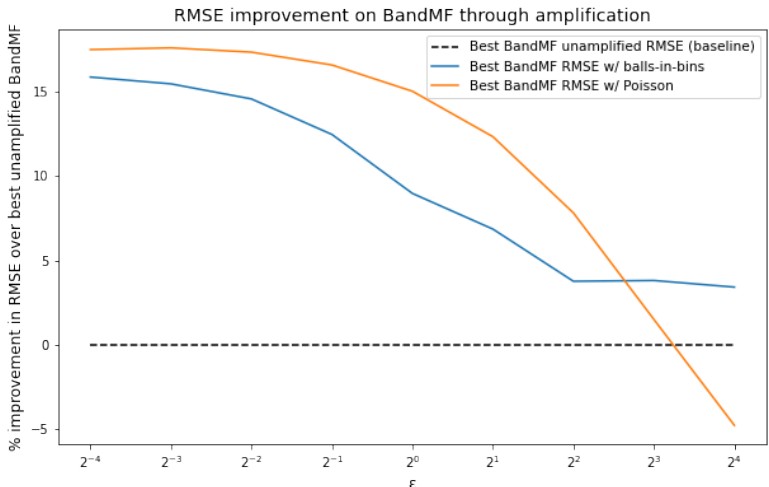

Figure 20: Improvement in RMSE over unamplified $b$-banded matrix factorization due to different amplification schemes. All curves optimize the choice of $b$ at each point.

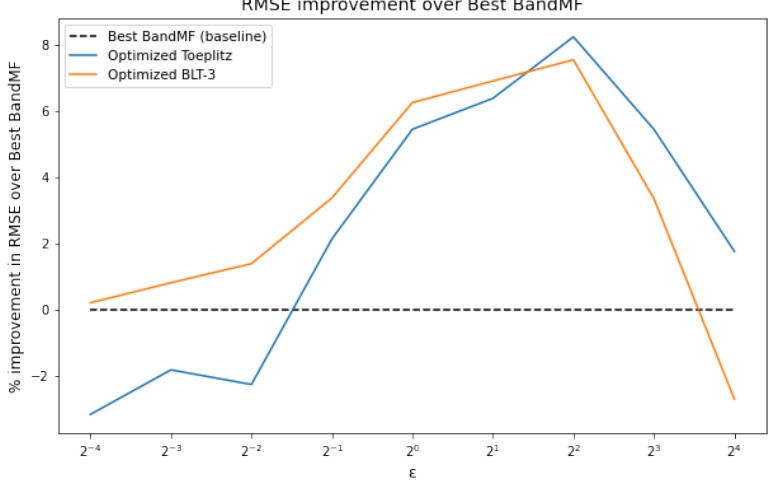

Figure 21: Improvement from optimizing $\mathbf{C}$ using objective accounting for amplification.

### F.11 Number of epochs = 32 and iterations per epoch = 128

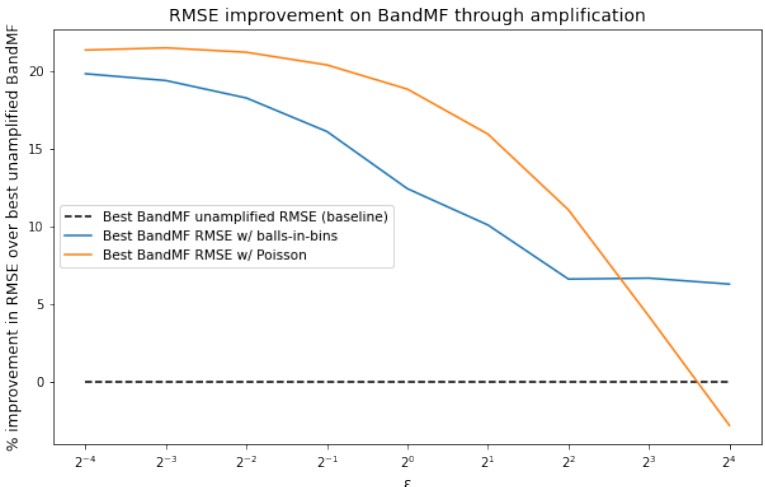

Figure 22: Improvement in RMSE over unamplified $b$-banded matrix factorization due to different amplification schemes. All curves optimize the choice of $b$ at each point.

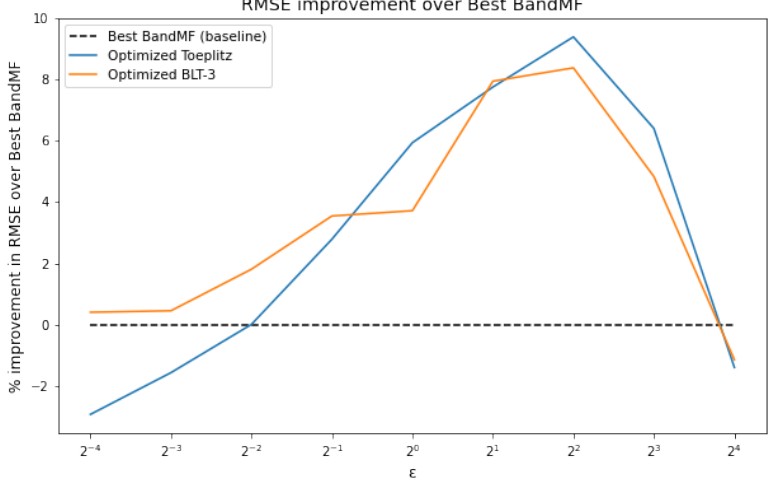

Figure 23: Improvement from optimizing $\mathbf{C}$ using objective accounting for amplification.

| Number of Samples = 131072 | | | | |
|---|---|---|---|---|
| | | Number of Epochs | | |
| | | 16 | 32 | 64 |
| Iterations per epoch | 32 | 376.88 | 384.44 | 311.55 |
| | 64 | 383.19 | 323.56 | 658.95 |
| | 128 | 389.37 | 770.18 | 3594.55 |

| Number of Samples = 1048576 | | | | |
|---|---|---|---|---|
| | | Number of Epochs | | |
| | | 16 | 32 | 64 |
| Iterations per epoch | 32 | 374.33 | 394.50 | 468.48 |
| | 64 | 525.25 | 664.23 | 804.56 |
| | 128 | 1060.45 | 1435.47 | Out of memory |

Figure 24: Time (in seconds) to optimize $\mathbf{C}$ for different values of Monte Carlo samples used per gradient descent iteration, number of epochs, and iterations per epoch.

## G    SCALABILITY OF OUR OPTIMIZATION PROCEDURE

### G.1    TIMING ANALYSIS

In this section, to understand the scalability of our optimization procedure, we analyze how long it takes to optimize for the matrix $\mathbf{C}$ as well and how it depends on the various parameters.

In Figure 24, we give the time, in seconds, for completing 300 iterations of gradient descent while varying the number of epochs, iterations per epoch, and number of samples per gradient descent iteration. We use a v100 GPU to perform the gradient steps. Here we fix $\delta = 10^{-5}$ and $\varepsilon = 1$. We make the following observations:

- Increasing the number of samples multiplicatively by 8 does not cause the runtime to increase by the same amount.
- On average, the runtime seems to have a sublinear dependence in the number of epochs.
- In contrast, in most settings for a large enough number of iterations per epoch the runtime seems to grow linearly or worse in the iterations per epoch.

For the largest number of samples, epochs, and iterations per epoch we ran into memory issues. In the next section, we show that using a very small number of samples in the optimization loop still yields good RMSE, i.e. these memory issues can generally be avoided. Combined with these observations, we believe this is evidence the optimization procedure is scalable in the number of epochs, although it may be infeasible to run the procedure for a large number of iterations per epoch.

### G.2    EFFECT OF THE NUMBER OF SAMPLES

When optimizing over the matrix $\mathbf{C}$, we use a Monte Carlo sampler to estimate $\sigma$. However, for privacy purposes we only need the estimate of $\sigma$ to be accurate for the final $\mathbf{C}$ output by the optimization procedure, since that is the only correlation matrix we will actually use during model training. For intermediate values of $\mathbf{C}$ in the gradient descent procedure, it is not a privacy violation to compute inaccurate $\sigma$ values for these matrices.

In turn, to make the optimization procedure more efficient, we can use a smaller number of samples per iteration. A natural question is then if the number of samples needed for the Monte Carlo estimator to

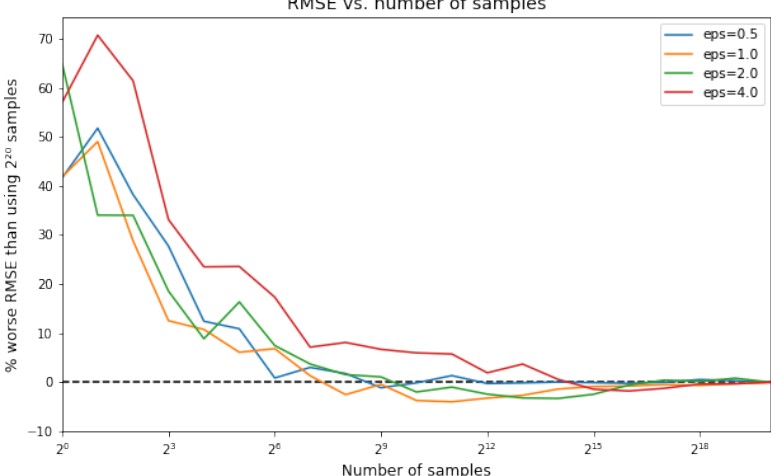

Figure 25: Percentage increase in RMSE due to using a smaller number of samples per iteration in the optimization procedure, when compared to the RMSE achieved by using $2^{20}$ samples per iteration.

concentrate is comparable to the number of samples needed for the optimization to achieve reasonable RMSE.

We run our optimization procedure for a varying number of samples per iteration, and in Fig. 25 for each of these numbers we plot the suboptimality of the final $\mathbf{C}$ value in terms of RMSE achieved compared to using the maximum number of samples. We vary $\varepsilon$ and fix $\delta = 10^{-5}$. For this choice of $\delta$, we need at least $1/\delta \approx 2^{17}$ samples are needed for the Monte Carlo estimator of $\delta$ to guarantee an error smaller than $\delta$ with, say, constant probability. However, in Figure 25, we see that for $\varepsilon = 0.5, 1, 2$ using e.g. $\approx 2^9$ samples per iteration suffices for achieving similar RMSE as $2^{20}$ samples per iteration, and at $\varepsilon = 4$ this number of samples only leads to a $\approx 10\%$ increase in RMSE. In other words, our optimization procedure can use a much smaller number of samples (compared to the final verification of $\mathbf{C}$ and $\sigma$) at little to no cost in utility.

## H    ADDITIONAL FIGURES FOR SECTION 5

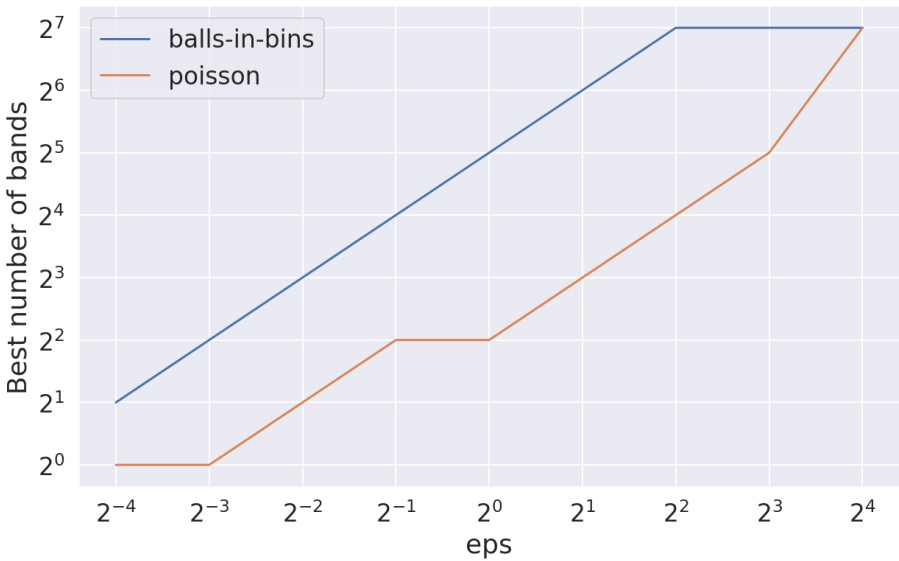

Figure 26: For each of the amplification methods in Figure 3a, the best choice of $b$ at each value of $\varepsilon$.

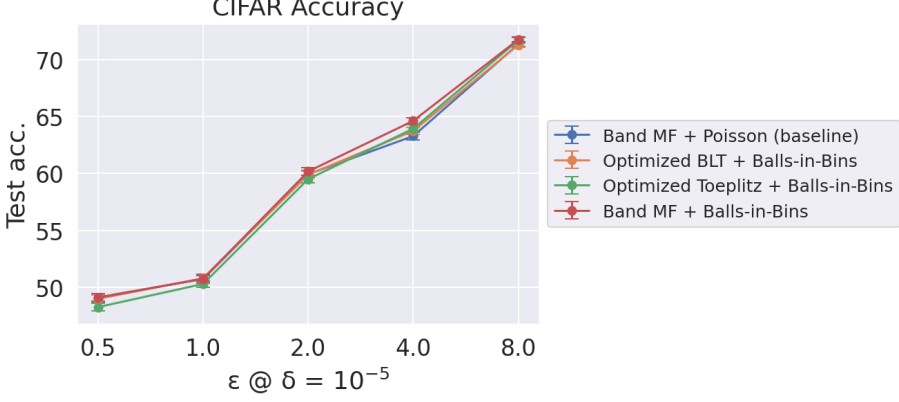

Figure 27: Comparison of accuracy on CIFAR10 of different correlated noise strategies and amplification methods, with 95% confidence intervals over 100 trials.

| Method / $\varepsilon$ | 0.5 | 1.0 | 2.0 | 4.0 | 8.0 |
|---|---|---|---|---|---|
| BandMF + Poisson | 1.018 | 0.778 | 0.606 | 0.481 | 0.388 |
| BandMF + Balls-in-Bins | 2.829 | 2.071 | 1.455 | 0.802 | 0.470 |
| Optimized BLT + Balls-in-Bins | 2.609 | 1.598 | 1.079 | 0.694 | 0.448 |
| Optimized Toeplitz + Balls-in-Bins | 2.003 | 1.182 | 0.864 | 0.639 | 0.445 |

Figure 28: Noise multiplier $\sigma$ used in the CIFAR10 experiments for different correlated noise strategies and amplification methods. Note that different strategies which have higher noise cancellation require higher $\sigma$, so a lower noise multiplier alone does not imply better learning performance.

| Method / $\varepsilon$ | 0.5 | 1.0 | 2.0 | 4.0 | 8.0 |
|---|---|---|---|---|---|
| BandMF + Poisson | 2 | 4 | 8 | 16 | 32 |
| BandMF + Balls-in-Bins | 16 | 32 | 64 | 64 | 64 |

Figure 29: Number of bands used in the CIFAR10 experiments for BandMF combined with different amplification methods.

## I   PRACTICAL IMPLEMENTATION OF BALLS-IN-BINS BATCHING

In Section 5.2, and in most empirical research on DP model training, an amplification method like Poisson sampling is assumed when computing the noise multiplier, but for convenience a simpler sampling scheme is actually implemented. The most common case of this is assuming Poisson sampling when instead doing shuffling, even though shuffling can produce much larger $\varepsilon$ values than sampling (Chua et al., 2024). A notable exception is research built on Opacus (Yousefpour et al., 2021) which implements Poisson sampling.

The main issue with implementing such sampling schemes properly is that they usually lead to unequal batch sizes (as schemes like shuffling which enforce equal batch sizes are usually hard to analyze tightly), which can lead to several inefficiencies:

- Modern model training pipelines are optimized for a fixed batch size. For example, XLA, which is used to e.g. compile gradient computations implemented in libraries like TensorFlow and JAX, requires a static input shape. To use variable batch sizes, one would naively need to forgo the use of XLA which could heavily slow down training in large-scale settings.
    - For example, consider Opacus which correctly implements Poisson sampling and supports variable batch sizes. In the benchmarking experiments of (Yousefpour et al., 2021), Opacus was shown to have runtime competitive with JAX for small-scale DP training. However, they observed that e.g. JAX pays a large up-front cost for compilation but after compilation achieves better runtime per iteration than Opacus. We expect that for larger scale the training the time spent on compilation is a smaller fraction of training time, hence variable batch size training with Opacus would be less scalable than XLA-based training using fixed batch sizes.
- Sampling a slightly larger batch can lead to disproportionately longer gradient computation times. As an extreme case, suppose we use an expected batch size of $B$, and we have $B$ accelerators that can each process a single gradient in parallel in time $T$. Hence computing a batch gradient on $B$ examples takes time $T$. However, if we have $B + 1$ samples instead, we will need time $2T$ instead.
- Sampling a smaller batch leads to wasted computational resources. For example, in the above setting, if we sampled $9B/10$ examples, then we would have $B/10$ accelerators being unused, which may be undesirable.

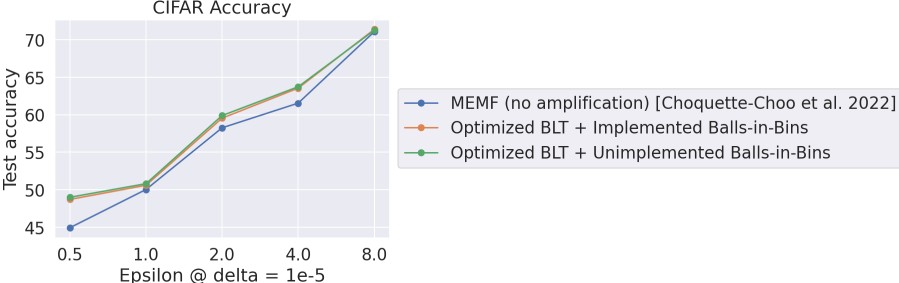

Figure 30: Comparison of implementing balls-in-bins batching to shuffling and assuming balls-in-bins in the analysis, and the unamplified baseline.

To this end we propose and implement the following practical variant of balls-in-bins batching: We do balls-in-bins batching, but:

- If a batch has size less than $B$, we pad it with some examples that have gradient 0 so its batch size becomes $B$.
- If a batch has size greater than $B$, we truncate it to have size exactly $B$.
- We compute the average gradient over the padded/truncated batch (i.e. our normalization in the average is always $1/B$).

Recall that balls-in-bins (without padding and truncating) can be implemented by shuffling the dataset, sampling a batch size schedule, and then taking batches according to this schedule. Hence even with padding and truncating balls-in-bins requires minimal overhead on top of shuffling, and indeed we were easily able to implement it in our CIFAR training code using only a single shuffle on the dataset and `numpy`'s multinomial sampler and pad operations, and use XLA to compile gradient computations in the resulting code. Furthermore, none of these changes affects the privacy analysis. This is because (i) the fixed normalization is compatible with the privacy analysis which is implicitly analyzing at the sum rather than the average, (ii) truncating a batch is equivalent to some of the examples in that batch having gradient 0, and DP-SGD's privacy analysis allows arbitrary clipped gradient.

We next demonstrate that the loss in utility due to truncating examples or not saturating the batch size in every iteration is acceptable. We rerun the training procedure from the previous section using BLT matrices (we focus on BLT matrices as they are the most scalable choice of correlated noise (McMahan et al., 2024)), but implement our practical variant of balls-in-bins batching. We pad/truncate to the same batch size of 500, i.e. in terms of gradient computations our practical variant of balls-in-bins is no more costly than shuffling the dataset and using fixed batch sizes instead.

In Figure 30 we compare this implementation to (i) the same implementation but using shuffling instead of balls-in-bins batching, and (ii) the state-of-the-art unamplified baseline of MEMF. We see that *the improvements from amplification over the unamplified baseline are far larger than the loss in utility due to the suboptimal batching introduced by the sampling scheme*. To the best of our knowledge, this is the first DP model training result that simultaneously (i) implements the sampling scheme assumed when computing the noise multiplier, (ii) does so at a negligible cost in efficiency and in a manner compatible with modern machine learning frameworks, and (iii) still demonstrates improvements over the unamplified baseline.

## J  DIMENSIONALITY REDUCTION AND ADAPTIVITY

To find a dominating pair for a matrix $\mathbf{C}$, we typically wish to argue that the worst case PLD is generated by a user that only ever inputs the same entry every iteration. This simultaneously handles dimension reduction and adaptivity and is the strategy for unamplified privacy analyses (see, e.g., Choquette-Choo et al. (2022)). In the MEMF case, for it to be the case that the worst-case PLD is achieved by a user always outputting the same vector, it suffices to require $\mathbf{C}_{:,i} \cdot \mathbf{C}_{:,j} \geq 0$ for all indices $i, j$ for which a user can simultaneously appear in. However, this is no longer the case for balls-in-bins sampling, even in a very simple case. Consider the matrix

$$\mathbf{C} = \begin{pmatrix} 1 & 0 \\ -1 & 1 \end{pmatrix}$$

with 1 epoch. Then, the user only participates in exactly one of the first column or the second. Say the user's inputs are $x = (a, b)$, where $a, b \in \{-1, 1\}$, which means that the MoG representing it

$$\frac{1}{2}\mathcal{N}((a, -a), \sigma^2 I) + \frac{1}{2}\mathcal{N}((0, b), \sigma^2 I).$$

Then, maximizing the inner product between $(a, -a)$ and $(0, b)$ will dominate the PLD w.r.t. $N(0, \sigma^2 I)$. This is then maximized at $b = -a$, rather than $b = a$. Thus, for balls-in-bins sampling, the inner products across columns that a user cannot simultaneously participate in matters. Noting that this example hinged on the inner product between the two columns being negative, it is possible that $\mathbf{C}^T \mathbf{C} \geq 0$ entry-wise is sufficient to show that a user may as well only input the same vector.

