# OpenReview forum: "Near-Exact Privacy Amplification for Matrix Mechanisms"
_ICLR.cc/2025/Conference — ICLR 2025 Poster_

### Official Review · Reviewer_kfPY · 2024-11-02

**Soundness:** 2
**Presentation:** 2
**Contribution:** 2
**Rating:** 5
**Confidence:** 4

**Summary:**

This paper presents an algorithm for calculating privacy parameters in differentially private SGD when using privacy amplification through random batching and noise correlated across rounds via a correlation matrix, known as the matrix mechanism.

**Strengths:**

This paper studies a very important problem, balls-in-bins batching with correlated noise, an alternative privacy amplification method in DP-SGD different from the classic subsampling amplification. Existing methods for analyzing privacy in this setting have drawbacks, especially with specific types of correlation matrices (banded matrices). The simulation method proposed shows an interesting and simple way to estimate the security parameter for arbitrary correlation matrix, which can be further used to optimize the noise form.

**Weaknesses:**

1. A significant limitation of the results is that, while a concrete simulation algorithm is presented, the paper lacks formal high-probability or confidence bounds for the computed DP security parameter. This absence leaves ambiguity about the paper's contribution: whether it offers merely a heuristic estimation or an approach with provable guarantees. My evaluation could improve if high-confidence bounds were provided.

2. The paper does not provide a closed-form or asymptotic bound for the amplification rate produced by the methods, or, equivalently, an indication of the amount of noise required or potentially saved. This lack of theoretical clarity makes it difficult to determine the extent of improvement the method could yield beyond empirical optimization of the noise form.

**Questions:**

1. How to derive high-confidence bound for the monte-carlo simulation results?

2. Can the authors provide the noise parameter or the noise variance in Figure 4 to help readers understand the amplification? The accuracy itself does not tell too much meaningful information as the performance of SGD may be influenced by many other factors.

3. Can the authors comment more on why in large eps, the proposed method performs much worse compared to subsampling amplification?

---

> ### Author Response · Authors · 2024-11-19
> **Response to Reviewer kfPY**
>
> Thanks to the reviewer for their feedback which has helped us improve the presentation of the paper. Below we respond to individual points raised by the reviewer:
>
> **How to derive high-confidence bound for the monte-carlo simulation results?**
> * For concentration we used Bernstein’s inequality in conjunction with the Wang et al. result. We added Theorem 2.2 and a corresponding appendix to explain this in more detail. In the original submission we did use Theorem 2.2 to guarantee all our empirical results satisfied the desired privacy guarantee via the wrapper of Wang et al. but regretfully neglected to discuss the details. We now emphasize this point in the paper at the start of Section 5.
>
> **The paper does not provide a closed-form or asymptotic bound for the amplification rate**
> * While we agree theoretical bounds on the improvements would be useful for a better understanding of amplification, we feel this is a challenge common to most privacy amplification results and not a major weakness, nor one specific to our paper. For example, even in the most idealized setting of Poisson sampling with independent noise and targeting RDP instead of (epsilon, delta)-DP bounds, the best theoretical results of Mironov et al. (https://arxiv.org/abs/1908.10530) require somewhat complex conditions that only hold when targeting small epsilons, and they still eventually show that approximate numerical computation is much better than their bounds in practice. Correlated noise heavily complicates the problem of computing privacy bounds under amplification for reasons we discuss in the paper, hence we believe it would both be extremely challenging to arrive at any non-trivial theoretical bounds, and simultaneously not very predictive of what we get by doing accounting via numerical methods.
>
> **Can the authors provide the noise parameter or the noise variance in Figure 4**
> * We have added this as a figure in the appendix, “Additional CIFAR Plots”. However, we note that because correlated noise mechanisms with larger noise cancellation require higher noise multipliers, comparing the noise multipliers of different methods is not particularly informative (even when comparing the results for Band MF, as the optimal banded matrix heavily differs between the two amplification methods). The RMSE metric, which is studied extensively in the previous subsection, is a better answer to the reviewer’s question because it accounts for both the noise multiplier and amount of noise cancellation.
>
> **Can the authors comment more on why in large eps, the proposed method performs much worse compared to subsampling amplification?**
> * We believe the reviewer is referring to the setting in Figure 2a, where independent noise is used instead of correlated noise; in all other settings, our method actually is comparable to or better than the result of Choquette-Choo et al. at higher epsilons.
> * The intuition for the gap in Figure 2a: Poisson sampling is strictly more random than balls-in-bins sampling (balls-in-bins can be viewed as Poisson sampling conditioned on a single participation per epoch). Since privacy amplification intuitively compounds sampling randomness with noise randomness, we expect Poisson sampling to provide better amplification than balls-in-bins sampling in most settings. This explains the gap in Figure 2a, where independent noise is used.
> * The reason this gap doesn’t manifest in other settings is that when we use correlated noise, we defer to the less random version of Poisson sampling proposed by Choquette-Choo et al. that in turn provides less amplification, whereas the amount of randomness in balls-in-bins is the same in all settings in the paper. If we could efficiently compute tight privacy guarantees for “maximally random” Poisson sampling with correlated noise, it’s likely that it would outperform balls-in-bins in most settings. However, an efficient and tight analysis is beyond our current understanding.

---

> ### Author Response · Authors · 2024-11-25
> **Thanks again for feedback**
>
> Thanks again to the reviewer for your feedback on the paper. Since the discussion period is ending soon, we would appreciate any thoughts from the reviewer on the rebuttal/revisions and whether there are any remaining concerns we could try to address.

---

> > ### Comment · Reviewer_kfPY · 2024-11-25
> >
> > Thanks for the authors' detailed response and I have increased my score. I agree that closed-form amplification could be challenging but it would be more helpful to show the comparison between matrix mechanism and subsampling amplification across a wider range of subsampling/partition rate, especially for the low rate scenario. It would be interesting to show if  matrix mechanism can lead to better performance in the low rate case which subsampling amplification is known to suffer from (e.g. see "Subsampled rényi differential privacy and analytical moments accountant" and "Improving the Gaussian Mechanism for Differential Privacy: Analytical Calibration and Optimal Denoising").

---

> > > ### Author Response · Authors · 2024-11-26
> > > **Added figure**
> > >
> > > Thanks to the reviewer for their response and the additional feedback. We agree a comparison of how the sampling probability / ratio of batch size to dataset size (which is what we believe the reviewer means by 'low rate' - please correct us if this is the wrong interpretation) affects the two sampling schemes is important to include. We plan to include a detailed empirical study of this comparison. We have added Figure 5 in Appendix F.1 as an example of the comparisons we plan to include. To summarize: Poisson sampling of Choquette-Choo et al. does better at higher sampling probabilities, but at lower sampling probabilities balls-in-bins does better (we believe because of the slack in the sampling scheme of Choquette-Choo et al.)
> > >
> > > Since we generated this figure in a short amount of time, we did not get to generate many samples for Monte Carlo verification and try a wider range of epsilons and sampling probabilities. We plan to include in the camera-ready version Figure 5 using more samples for Monte Carlo verification, and reproduce Figure 5 for other settings of epsilons and a sweep of the other parameters.

---

> > > > ### Comment · Reviewer_kfPY · 2024-11-30
> > > > **On its application to SGD**
> > > >
> > > > Thank you for your reply. I have one more major technical question. When I tried to understand the application of the matrix mechanism to DP-SGD, I am unsure whether the static linear aggregation model studied in this paper is directly applicable. This paper claims that the problem can be reduced to releasing \( Cx + z \), where \( C \) is a correlation matrix, \( x \) represents the gradient used for updates, and \( z \) is Gaussian noise.
> > > >
> > > > Based on my understanding, the approach assumes that \( x \) is a predetermined matrix, where each datapoint influences a single row. The method involves searching for and optimizing the selection of \( C \) to ensure that the required noise satisfies privacy constraints while minimizing the RMSE.
> > > >
> > > > My main concern is related to the privacy analysis. In SGD, \( x \) is not a predetermined matrix. Consider a simple two-iteration scenario where \( x = (x_1; x_2) \). Here, \( x_2 \) is determined by the **noisy** iterate after updating \( x_1 \), along with additional noise. This behavior differs from the sensitivity analysis of \( Cx \), i.e., \( C(x - x') \), for two adjacent matrices \( x \) and \( x' \) differing in a single row.
> > > >
> > > > Continuing with the two-iteration example, suppose the differing datapoint in the adjacent datasets appears in the first iteration. In this case, $\|x_1 - x'_1\|$ can be as large as the clipping threshold \( c \), and \( x_2 \) may also differ from \( x'_2 \) since their preceding noisy iterates influence them. While I can see a potential analysis using composition, it remains unclear how to rigorously study the sensitivity of \( Cx \) in the context of DP-SGD.

---

> > > > > ### Author Response · Authors · 2024-12-01
> > > > > **Re: On its application to SGD**
> > > > >
> > > > > We thank the reviewer for this point. We do take the setting to be adaptive queries  and our work builds on the prior work on connecting matrix mechanisms to DP-SGD. In particular, [Denisov et al. (2022, Theorem 2.1)](https://arxiv.org/pdf/2202.08312) shows that privacy guarantees for standard DP-SGD in the adaptive case are the same as in the non-adaptive case (this is closely related to the filter property of Gaussian mechanism ([https://arxiv.org/abs/2210.17520](https://arxiv.org/abs/2210.17520)) and [Choquette-Choo et al. (2023, Theorem 5)](https://arxiv.org/pdf/2306.08153) extend this to the setting with amplification due to Poisson subsampling.
> > > > >
> > > > > The reviewer's primary concern is that we justify the claim that we can assume all non-sensitive gradients are zero by subtracting out $Cx'$ from both distributions; however, as the reviewer notes, this is complicated in the adaptive setting. We agree that this argument isn't detailed enough for the adaptive case and we will update our submission to clarify and elaborate our argument. Our more precise argument is a simple simulation argument which shows that the adversary can simulate the output on a general dataset from the output on the dataset with non-sensitive gradients are zero. Crucially, we require that $C$ is lower triangular and the masking process is independent across data samples (which holds for our balls-in-bins sampling as well as Poisson sampling). To be more precise, we sketch the argument below (similar to in Choquette-Choo et al. (2023)):
> > > > > * We first introduce the following notation:
> > > > > 	* Let $D$ be a dataset of $n$ elements and $D'$ be an adjacent dataset with the first element removed.
> > > > > 	* For all $i,j \in [n]$, let $x^{(i)}_j : \mathbb{R}^{j-1} \rightarrow \mathbb{R}^p$ be the adaptive gradient at iteration $j$ for user $i$ (represented here as a function that takes as input the previous mechanism outputs)
> > > > > 	* Let $P^{(i)} \in \{0,1\}^{n \times n}$ be the random mask applied to this user's gradients.
> > > > > 	* Let $O = C \sum_{i=1}^n P^{(i)} x^{(i)} + z$ be the output of the matrix mechanism on the whole dataset and $O' = C \sum_{i=2}^n P^{(i)} x'^{(i)} + z$ be the output with the first user removed, where $x^{(i)}\_j$ (resp. $x'^{(i)}\_j$) implicitly takes in $O\_{1:j-1}$ (resp. $O'\_{1:j-1}$) as input.
> > > > > 	* Also $A = C P^{(1)} g + z$ and $A' = z$ where $g$ is also an adaptive gradient taking in $A$ as input.
> > > > > * We wish to show that the privacy guarantee between $O$ and $O'$ is no worse than the privacy guarantee between $A$ and $A'$ for some choice of $g$.
> > > > > * Suppose we tell the adversary the values of $P^{(i)}$ for $i > 1$. This can only weaken our privacy guarantee between $O$ and $O'$ for arbitrary gradients and because the masks are independent across $i$, this doesn't give the adversary any additional information about $P^{(1)}$.
> > > > > * It can now be seen inductively that the adversary can adaptively simulate $O$ and $\{x^{(i)}\}$ from having draws from $A$ with a specific choice of $g$:
> > > > > 	* Given that $O\_{1:j-1}$ and $\{x^{(i)}\_{1:j-1}\}$ have already been simulated,  $\{x^{(i)}\_j\}$ is immediately generated by passing in $O\_{1:j-1}$.
> > > > > 	* Then we choose $g\_j$ to be $x^{(1)}\_j$ (which can be seen as a function of $A\_{1:j-1}$).
> > > > > 	* With this choice, we can then obtain $O_{1:j}$ by taking $A_{1:j} + C_{1:j,:}\sum_{i=2}^n P^{(i)}x^{(i)}$ (where the fact that $C$ is lower triangular implies that the second term only depends on $x_{1:j}^{(i)}$, which is already known, thus preserving adaptivity)
> > > > > 	* This proves the claim.
> > > > > * One can similarly show that the identical process with $g=0$ simulates $O'$ from $A'$.
> > > > > * Because an identical process transforms $A$ to $O$ and $A'$ to $O'$, it follows from post-processing that the privacy between $O$ and $O'$ is no worse than the privacy between $A$ and $A'$.

---

> > > > > > ### Author Response · Authors · 2024-12-02
> > > > > >
> > > > > > Thanks again to the reviewer for their time and thoughtful comments on the paper. As the discussion period is closing soon, we were wondering if this response adequately addresses your concerns about the applicability of our work in the DP-SGD setting or if you still have any lingering questions.

---

> > > > > > > ### Comment · Reviewer_kfPY · 2024-12-02
> > > > > > >
> > > > > > > Thank you very much for the response, which has fully addressed my concern.

---

### Official Review · Reviewer_6FLQ · 2024-11-04

**Soundness:** 3
**Presentation:** 3
**Contribution:** 3
**Rating:** 8
**Confidence:** 3

**Summary:**

The paper provides a differential privacy accountant for the "ball-in-bins batching", where the disjoint mini-batches for DP-SGD are formed via multinomial sampling. A particularly nice property of the accounting method is that it can be combined with so-called matrix mechanisms, where correlated noise is added to the DP-SGD iterations. The accounting is based on the randomized differential privacy accounting framework provided by [Wang et al., 2023](https://proceedings.neurips.cc/paper_files/paper/2023/file/6ae7df1f40f5faeda474b36b61197822-Paper-Conference.pdf), where Monte Carle sampling of the hockey-stick integral (when expressed using the privacy loss distribution) is carried out. This is enabled by Lemma 3.2 which gives a dominating pair of distributions $(P,Q)$, where $P$ can be sampled efficiently and $Q$ has a tractable density function. This way, the PLD can be sampled and the hockey-stick integral estimated.

**Strengths:**

- The paper provides a novel and practical mini-batch sampling scheme applicable to matrix mechanisms.

- Seems like a novel way for using the randomized accounting framework proposed by Wang et al. (2023). To the best of my knowledge, this is the most "non-trivial" use of that framework and sets an example that the Monte Carlo sampling is viable tool for differential privacy accounting and designing DP mechanisms.

- Although the convergence analysis is missing, and there are many future questions (e.g., the tightness of the dominating pair and the efficiency of the sampling), the paper gives convincing experimental results indicating this is a viable way of carrying out privacy accounting for the proposed mini-batch sampling scheme.

**Weaknesses:**

- The theory is a weak side of the paper. There are really no theoretical guarantees given for the accuracy of the estimates. Theorem 2.1 is stated (Theorem 9 by Wang et al., 2023) however there are no results that would materialize this result. Also, I think it would be good to elaborate on that result, especially since it is stated so differently than Theorem 9 by Wang et al.

- The paper is not fully polished: I don't see the neighborhood relation of datasets stated anywhere. Due to the form of the dominating pair $(P,Q)$ given in Section 3, I assume you consider the add/remove relation. Perhaps the definition you use could be added somewhere in Section 2?

**Questions:**

- In Lemma 3.2: What exactly guarantees that $(P,Q)$ dominates the pair $(Q,P)$ (for those stated $P$ and $Q$) ? I think the proof of Lemma 3.2 could be elaborated to make it more readable. Perhaps you could add the required results from the reference paper (Choquette-Choo et al., 2024b) to the appendix instead of shortly referring to the lemmas and theorems (example: "see e.g. the first part of proof of Theorem 4.8").

- What would be required to for providing convergence guarantees for the estimates? Could you use the results by Wang et al. for that?

- As I said I assume the analysis is for the add/remove neighborhood relation of datasets. In addition to adding it somewhere (e.g., Section 2), you could briefly discuss in Section 3 how the results might change (or not) for the substitution neighborhood relation.

---

> ### Author Response · Authors · 2024-11-19
> **Response to Reviewer 6FLQ**
>
> Thanks to the reviewer for their questions which have helped us improve the clarity of the paper. We address below individual questions/weaknesses raised by the reviewer:
>
> **The paper is not fully polished: I don't see the neighborhood relation of datasets stated anywhere.**
> * This is a good point. Yes, we are using add/remove. We now make this clear in Section 2 and Lemma 3.2.
>
> **What exactly guarantees that (P, Q) dominates the pair (Q, P)**
> * We now state explicitly in Lemma 3.2 that (P, Q) is the dominating pair for the add adjacency, and (Q, P) is the dominating pair for the remove adjacency. (P, Q) does not necessarily dominate (Q, P), but we can do still accounting for add-remove by taking the worse of the DP guarantees calculated using the two dominating pairs, and by a union bound this at most doubles the failure probability we use in Theorem 2.1 (this is discussed in Appendix E).
>
> **I think the proof of Lemma 3.2 could be elaborated to make it more readable.**
> * We have made the proof more thorough and self-contained.
>
> **What would be required to for providing convergence guarantees for the estimates?**
> * For concentration we used Bernstein’s inequality in conjunction with the Wang et al. result. We added Theorem 2.2 and a corresponding appendix to explain this in more detail. In the original submission we did use Theorem 2.2 to guarantee all our empirical results satisfied the desired privacy guarantee via the wrapper of Wang et al. but regretfully neglected to discuss the details. We now emphasize this point in the paper at the start of Section 5.
>
> **How the results might change (or not) for the substitution neighborhood relation.**
> * The recent work of Schuchardt et al. (https://arxiv.org/abs/2403.04867) allows us to reprove Lemma 3.2 for the substitution adjacency by basically the same proof. It is straightforward to adapt our Monte Carlo sampling code to the resulting pair of dominating distributions, since this only changes the computation of the privacy loss/log likelihood ratio. We will include a discussion of this in the paper.

---

> ### Comment · Reviewer_6FLQ · 2024-11-24
>
> Thank you for the replies. After these clarifications and modifications to the paper, I am raising my score.

---

### Official Review · Reviewer_7a3D · 2024-11-06

**Soundness:** 3
**Presentation:** 3
**Contribution:** 3
**Rating:** 6
**Confidence:** 3

**Summary:**

In this paper, the authors propose a ``balls-and-bins'' batching scheme, for which the privacy analysis is quite straightforward (by analyzing hockey-stick divergence of the `dominating pairs'). Based on their novel batching scheme, the authors use Monte Carlo accounting to estimate the privacy parameter and avoid the estimation errors introduced by DP composition theorems. The experimental results indicate that their scheme performs better than existing works.

**Strengths:**

The paper is well-written, and the theoretical results seem to be correct (I didn't check the full proof carefully, but I believe it makes sense at a high level).

**Weaknesses:**

- In the abstract, the authors claim that the ``ball-and-bins'' scheme is closer to practical random batching than Poisson sampling. But there is no theoretical explanation in the paper. Not sure if I miss anything here.
- The Algorithm 2 applies bisection method to compute the $\sigma$. But the proof of monotonicity is missing.

**Questions:**

- What is the ``practical random batching'', and why is the balls-and-bins scheme approximates it better than Poisson sampling?
- Can the authors provide a proof for the monotonicity between $\sigma$ and $\delta$?

---

> ### Author Response · Authors · 2024-11-19
> **Response to Reviewer 7a3D**
>
> Thanks to the reviewer for their questions which have helped us improve the clarity of the paper. Below we respond to individual weaknesses/questions raised by the reviewer:
>
> **What is the ``practical random batching'', and why is the balls-and-bins scheme approximates it better than Poisson sampling?**
> * We apologize for the lack of clarity. The practical random batching we refer to is just shuffling. We now say shuffling explicitly in the abstract. By “closer”, we mean the amount of overhead compared to shuffling needed to implement balls-in-bins is minimal. e.g., the paragraph “Advantages of balls-in-bins batching” explains how balls-in-bins batching can be implemented by shuffling and then using a variable batch size (as we do in Appendix H).
>
> **Proof of monotonicity is missing**
> * Since we just need to find an (approximate) solution and $\hat{\delta}$ is continuous in $\sigma$, monotonicity is not required for bisection to converge (and perhaps surprisingly, we have found counterexamples for monotonicity of $\hat{\delta}$). Instead, it suffices to have $\sigma_1$, $\sigma_2$ such that $\hat{\delta}(\sigma_1) > \delta$ and $\hat{\delta}(\sigma_2) < \delta$. We have added a discussion of this as a footnote when discussion bisection.
> * In more detail: $\hat{\delta}(\sigma_1) = 1$ for $\sigma_1 = 0$, and if we take enough samples then whp $\hat{\delta}(\sigma_2) < \delta$ for $\sigma_2$ set to the sigma achieving the target $(\epsilon, \tau \delta)$-DP guarantee without amplification.

---

> > ### Comment · Reviewer_7a3D · 2024-11-24
> >
> > Thanks for your reply. I maintain my score.

---

### Official Review · Reviewer_9yen · 2024-11-07

**Soundness:** 3
**Presentation:** 3
**Contribution:** 3
**Rating:** 6
**Confidence:** 3

**Summary:**

This paper investigates the computation of privacy parameters for differentially private (DP) machine learning, specifically when using privacy amplification through random batching and noise correlated across rounds via a correlation matrix C. Previous work either restricted C to banded matrices or provided loose privacy parameters. The authors introduce a framework for computing near-exact privacy parameters for any lower-triangular, non-negative C, optimizing C while considering amplification. They show empirical improvements over the state-of-the-art in root mean squared error (RMSE) for prefix sums and performance on deep learning tasks. The key techniques include Monte Carlo accounting to avoid composition and a balls-in-bins batching scheme for practical random batching.

**Strengths:**

- Propose a more practical minibatch sampling method, the Balls-in-Bins, that achieves better privacy amplification.

- Use Monte Carlo accounting to bypass composition. The limitations of the current work are well-discussed, especially the limitation of the construction of P.

**Weaknesses:**

See questions.

**Questions:**

- Could the authors interpret "near-exact" privacy analysis by:
  - Provide a more precise definition or explanation of what it means by "near-exact".
  - Discuss the potential impact of the looseness in $P$ on the overall privacy guarantees, given that the construction of $P$ in the dominating pair is potentially too loose.

- In lines 358-364, the authors discuss the choice of $b$ in experiments, which is chosen from the set {$ 1, 2, 4, 8, 16, 32, 64, 128, 256 $}. How does the (sub)optimal choice of $b$ vary with the other key parameters (e.g., $\varepsilon$)? Could the authors briefly discuss the intuition behind these variations, or include an ablation study showing the impact of suboptimal b choices on performance?

- The authors note that the "balls-in-bins” batching scheme enables easy privacy analysis. However, will the practical variant of balls-in-bins batching in Appendix H complicate the privacy analysis? Would it encounter the same difficulties in privacy analysis as Poisson sampling?

- In line 280, the authors mention that the bisection requires $O(\log(1/\delta))$ iterations, where $\delta$ is supposed to be the privacy parameter in $(\varepsilon, \delta)$-DP. However, should it be $O(\log(1/\text{err}))$, where $\text{err}$ is the error tolerance of $\sigma^*$?

Minor:
- Figure 1 explains the amplification methods quite well. However, I would suggest additional notes indicating that the first row corresponds to the batches in the first epoch (e.g., $E = 1$), and that the second row is the second epoch (e.g., $E = 2$).

- In the introduction, the authors mention that one of the limitations of previous work is the banded $C$ issue. Could the authors provide the key intuition as to why the banded $C$ may restrict utility and efficiency, or discuss any specific scenarios or applications where non-banded C matrices are particularly beneficial?

- In Appendix H, for the practical variant of balls-in-bins batching, the authors pad the small batch with some zero-gradient examples to avoid wasted computational resources. Could the authors further explain why wasted computational resources would be a problem?

---

> ### Author Response · Authors · 2024-11-19
> **Response to Reviewer 9yen**
>
> Thanks to the reviewer for their questions which have helped us improve the clarity of the paper. Below we respond to individual weaknesses/questions raised by the reviewer:
>
> **Could the authors interpret "near-exact" privacy analysis**
> * There are two potential sources of looseness: (i) looseness of the dominating pair and (ii) looseness in computing $D_\epsilon(P, Q)$ for the dominating pair. (ii) is merely a looseness in the MC-sampling procedure and can be made arbitrarily tight by increasing the number of samples (as well as runtime). (i) is tight in general: in the case where all examples have scalar gradient 0 except for the differing example, which has scalar gradient 1, the dominating pair (P, Q) is exactly the outputs of the corresponding matrix mechanism. We have added a clarification explaining (ii) to Section 1.1.1.
>
> **How does the (sub)optimal choice of $b$ vary**
> * Generally, as epsilon gets higher the number of bands increases. Effectively, lower bands gives better amplification and higher bands gives better noise cancellation. The former gets worse as the epsilon increases, whereas the latter is independent of noise, so the optimal point in this tradeoff skews towards higher numbers of bands as epsilon increases.
> * For comparing Poisson sampling of Choquette-Choo et al. to our balls-in-bins, balls-in-bins will generally use more bands because Choquette-Choo et al.’s sampling scheme uses a less random method, i.e. offers less amplification, as the number of bands increases, whereas balls-in-bins’ randomness does not change with the number of bands.
> * We added to the appendix “Additional Figures for Section 5” a figure showing how the optimal number of bands varies with epsilon for the two methods, which confirms the above intuition, and a pointer from Figure 3.a to this appendix.
>
> **Suggestions for Figure 1**
> * We have added a clarification to the caption that says each epoch is one row in the figure.
>
> **Will the practical variant of balls-in-bins batching in Appendix H complicate the privacy analysis?**
> * The privacy analysis of the practical balls-in-bins variant in App H is the same as that of the balls-in-bins we analyze! The zero-padding and trimming are both accounted for in the analysis of balls-in-bins - padding does not matter because our privacy analysis is effectively analyzing sum queries, and truncating is the same as some examples having zero gradient which is accounted for by our analysis.
>
> **Should it be $log(1/err)$?**
> * Yes, the reviewer is correct, thanks for pointing this out. We have fixed the error.
>
> **“Could the authors provide the key intuition as to why the banded may restrict utility and efficiency, or discuss any specific scenarios or applications where non-banded C matrices are particularly beneficial?”**
> * Effectively, the added memory/runtime requirement of using a b-banded matrix is linear in b, and to the best of our knowledge, to achieve within a constant factor of the utility of a full lower-triangular matrix requires b = Omega(n) (e.g. Theorem 6 of Kalinin and Lampert  https://arxiv.org/abs/2405.13763). So in general, one must choose between efficiency and utility when using banded matrices. In contrast Dvijotham et al. https://arxiv.org/abs/2404.16706 gives a non-banded matrix family that achieves (1+o(1)) times the optimal utility and the computational overhead is O(log n), which shows we can get the “best of both worlds” by choosing non-banded matrices.
>
> **Could the authors further explain why wasted computational resources would be a problem?**
> * The main reason we pad the batch is to keep the batch size fixed across iterations. This way, we only need to do XLA compilation of the gradient operation once across all of training, as opposed to once for each new batch size. However, the cost of this is we now spend part of our computation on these padding examples, whereas e.g. without DP / random batching we would spend all of our computation on actual examples.

---

> > ### Comment · Reviewer_9yen · 2024-11-24
> >
> > Thank you to the authors for their rebuttal. I have increased my rating accordingly.

---

### Meta-Review · Area_Chair_Ae17 · 2024-12-17

**Metareview:**

Reviewers agreed that the paper provides a novel and practical sampling scheme for DP-SGD, and appreciated the good empirical performance. Some technical concerns were addressed during the rebuttal phase. A common concern is that the paper lacks theoretical analysis/guarantee, though some reviewers are excited about the novel idea behind the proposed balls-in-bins mechanism and believed that this is the way to go for DP-SGD with disjoint batches. Overall, the sentiment appears to be positive.

**Additional Comments On Reviewer Discussion:**

See above. I kind of agreed with reviewer 6FLQ's reasoning that the paper can be a good spotlight paper. After reading everything again, I feel that if the paper provides the desired theoretical analysis, then I'd be more comfortable recommending spotlight or oral. I won't object to spotlight though.

---

### Decision · Program_Chairs · 2025-01-22

Accept (Poster)